# Single-cell transcriptomes of pancreatic preinvasive lesions and cancer reveal acinar metaplastic cells' heterogeneity

Yehuda Schlesinger[1,7], Oshri Yosefov-Levi[1,7], Dror Kolodkin-Gal[2,3,7], Roy Zvi Granit[1], Luriano Peters[1], Rachel Kalifa [2,3], Lei Xia[1], Abdelmajeed Nasereddin [4], Idit Shiff [4], Osher Amran[1], Yuval Nevo[5], Sharona Elgavish[5], Karine Atlan[6], Gideon Zamir [2✉] & Oren Parnas [1✉]

Acinar metaplasia is an initial step in a series of events that can lead to pancreatic cancer. Here we perform single-cell RNA-sequencing of mouse pancreas during the progression from preinvasive stages to tumor formation. Using a reporter gene, we identify metaplastic cells that originated from acinar cells and express two transcription factors, Onecut2 and Foxq1. Further analyses of metaplastic acinar cell heterogeneity define six acinar metaplastic cell types and states, including stomach-specific cell types. Localization of metaplastic cell types and mixture of different metaplastic cell types in the same pre-malignant lesion is shown. Finally, single-cell transcriptome analyses of tumor-associated stromal, immune, endothelial and fibroblast cells identify signals that may support tumor development, as well as the recruitment and education of immune cells. Our findings are consistent with the early, premalignant formation of an immunosuppressive environment mediated by interactions between acinar metaplastic cells and other cells in the microenvironment.

[1] The Concern Foundation Laboratories at the Lautenberg Center for Immunology and Cancer Research, IMRIC, Faculty of Medicine, Hebrew University–Hadassah Medical School, Jerusalem 91120, Israel. [2] Department of Surgery, Hadassah–Hebrew University Medical Center, Jerusalem 91120, Israel. [3] Department of Developmental Biology and Cancer Research, IMRIC, Faculty of Medicine, Hebrew University–Hadassah Medical School, Jerusalem 91120, Israel. [4] Genomics Applications Laboratory, Core Research Facility, Faculty of Medicine, The Hebrew University of Jerusalem, Jerusalem 91120, Israel. [5] Info-CORE, Bioinformatics Unit of the I-CORE at the Hebrew University of Jerusalem and Hadassah Medical Center, Jerusalem 91120, Israel. [6] Department of Pathology, Hadassah–Hebrew University Medical Center, Jerusalem 91120, Israel. [7] These authors contributed equally: Yehuda Schlesinger, Oshri Yosefov-Levi, Dror Kolodkin-Gal. ✉email: RgideonZ@hadassah.org.il; oren.parnas@gmail.com

Pancreatic ductal adenocarcinoma (PDAC) is a deadly cancer. The 5-year survival rate is around 10% with very little improvement over recent decades[1] and only a short delay in disease progression in response to available treatments. PDAC is believed to develop in a gradual manner over many years[2]. However, early detection is challenging[3] and late diagnosis is one of the major reasons for the poor prognosis and the high patient mortality. It is therefore important to achieve a better understanding of the early events that give rise to PDAC, and to explore the development of the disease from early preinvasive lesions to advanced tumor.

Lineage tracing studies in mice have suggested that one prominent route to PDAC is via acinar cells that undergo ductal metaplasia, followed by the formation of pancreatic intraepithelial neoplasia (PanIN) lesions and subsequently invasive cancer[4]. Histologically, the acinar cells' morphology changes and they form duct-like structures, but the molecular underpinnings of metaplasia are poorly understood, and little is known about heterogeneity among cells undergoing the process. Constitutively active mutant KRAS is a major early driver of PDAC[5] and is found in more than 90% of PDAC patients, as well as in human PanIN lesions[6,7]. Expression of constitutively active mutant Kras in acinar cells induces acinar-to-ductal metaplasia and low-grade PanINs in mice. The transition from low-grade PanINs to PDAC is accompanied by the accumulation of additional mutations in tumor suppressor genes and can be accelerated by environmental stresses, most prominently inflammation[8,9].

The cancer-protective, complex, multi cell-type tumor microenvironment is thought to be a major factor in the poor response of PDAC patients to treatment. Fibroblasts shape the microenvironment to create a desmoplastic reaction that reduces drug penetration, reduces the number of tumor-associated immune cells and induces tissue stiffness, making it harder to perform successful surgery[10–12]. In addition, fibroblasts secrete cytokines, such as IL-6[13], that alter the immune response. However, depletion of myofibroblasts (a subtype of activated fibroblasts) has been shown to reduce survival in an animal model[14,15].

Immune cells can be detected in pancreatic lesions and pancreatic tumors, but so far immunotherapy has not been shown to benefit PDAC patients[16], in part because of the dominant effect of suppressive cells in the tumor microenvironment (TME). Consistent with this possibility, depletion of myeloid cells, for example macrophages, reduces lesion formation and tumor volume in mice[17,18]. It is not known at which stage during the development of the disease the immunosuppressive environment is formed and what signals and cells are involved. The pro-tumorigenic and antitumorigenic effects of fibroblasts, immune cells and other stromal cells emphasize the need to accurately profile the different subpopulations and the interactions between cell types during the development of the disease.

Single-cell RNA-sequencing (scRNA-seq) is a powerful tool to profile complex organs and tumors containing multiple cell types that interact and signal to each other[19,20]. Previous studies using scRNA-seq of mouse pancreatic tumors or human PDAC samples gained new insight on this deadly disease[21–24]. However, a profile of the early events that alter acinar cells from normal to metaplastic and then to malignant cells, is missing and is critical to the understanding of PDAC development.

In the current study, we perform time course scRNA-seq experiments of mouse PDAC and human. The mouse model (inducible expression of Kras-G12D) is used to profile the changes in the stromal and acinar cells from preinvasive lesions to tumor. To accurately follow the changes in acinar cells and profile acinar metaplastic cells, we use genetically labeled acinar cells with tdTomato. Our data reveal the heterogeneity of acinar metaplastic cell types and their potential interactions with immune and stromal cells. These findings shed light on the sequence of events that lead to acinar cell transformation and reveal several metaplastic cell types and states, from which malignant cells can develop. In addition, we describe the transcriptional changes of fibroblast, endothelial cells and immune cells during the development of the disease and reveal new potential markers for early detection of PDAC.

## Results

**Pancreatic single-cell transcriptome**. To explore cell heterogeneity following the expression of constitutively active Kras in acinar cells, which causes acinar metaplasia and pancreatic dysplasia, we carried out scRNA-seq experiments of pancreatic tissues. Tamoxifen was injected into six- to eight-week-old Ptf1a-CreER, LSL-Kras-G12D, LSL-tdTomato (PRT) mice, and the pancreas was collected for single-cell isolation at six different time points post-tamoxifen injection (PTI) (Fig. 1a–h). Ductal structures and pancreatic intraepithelial neoplasia (PanIN), were rare in control, 17 days and 6 weeks PTI samples, but clearly accumulated starting at 3 months PTI (Supplementary Fig. 1a). Based on the number of PanIN lesions, we defined the control and two early time points, as early stage samples, while defining 3 months, 5 months, and 9 months PTI as late-stage samples. It is important to note that nearly all late-stage samples in this model, include low-grade PanINs, which are noninvasive. In addition, we also sampled a 15 months PTI mouse that developed an invasive PDAC. In total, 41,139 single cells from the pancreata of nine mice passed quality control criteria (see "Methods" section) and were included in the initial analysis presented in Fig. 1i, including acinar cells, ductal cells, fibroblasts, endothelial cells, neuroendocrine cells, pericytes, and immune cells (Supplementary Fig. 1b–i, Supplementary Data 1). Biological duplicate samples from both 3 and 5 months PTI mice were similar (Supplementary Fig. 1j, k), showing that batch effects were minimal. We notice several trends in the data: (i) In each cell type, cells from tissues taken at early time points PTI clustered together, and those from tissues taken at later time points PTI clustered together (Fig. 1i). Thus, although mutated Kras was expressed in acinar cells, the transcriptional profile of each cell type in the stroma changed, and these changes dominated transcriptional heterogeneity within each cell type. (ii) At late time points, the formation of PanIN lesions was accompanied by increased infiltration of immune cells (Fig. 1i–m). This was associated with the expression of pro-inflammatory genes in both epithelial and stromal cells (Supplementary Fig. 1o). (iii) At late time points PTI, we have identified tdTomato-positive cells that expressed Krt19 and Sox9 but did not express Cpa1, and other genes that encode exocrine enzymes (Fig. 2c–f and Supplementary Fig. 1l–n). We therefore conclude that these cells are metaplastic acinar cells that lost the acinar program. These acinar metaplastic cells can develop to PDAC and our setting allows us to capture their transcription profile and heterogeneity throughout the pathogenesis.

We next explored transcriptional changes in specific lineages along the development of the lesions, including epithelial cells, immune cells, endothelial cells, and fibroblasts.

**Acinar cells recovered from late time point lose identity**. We took advantage of the LSL-tdTomato reporter, which genetically marks acinar cells and their metaplastic progeny and clustered the acinar, ductal, and tdTomato positive cells based on the analysis mentioned above (Fig. 2a–e). The following phenotypes thus discriminate three types of epithelial cells: Acinar cells (Cpa1+, tdTomato+, Krt19−), ductal cells (Cpa1−, tdTomato−, Krt19+) and metaplastic cells (Cpa1−, tdTomato+, Krt19+). Therefore,

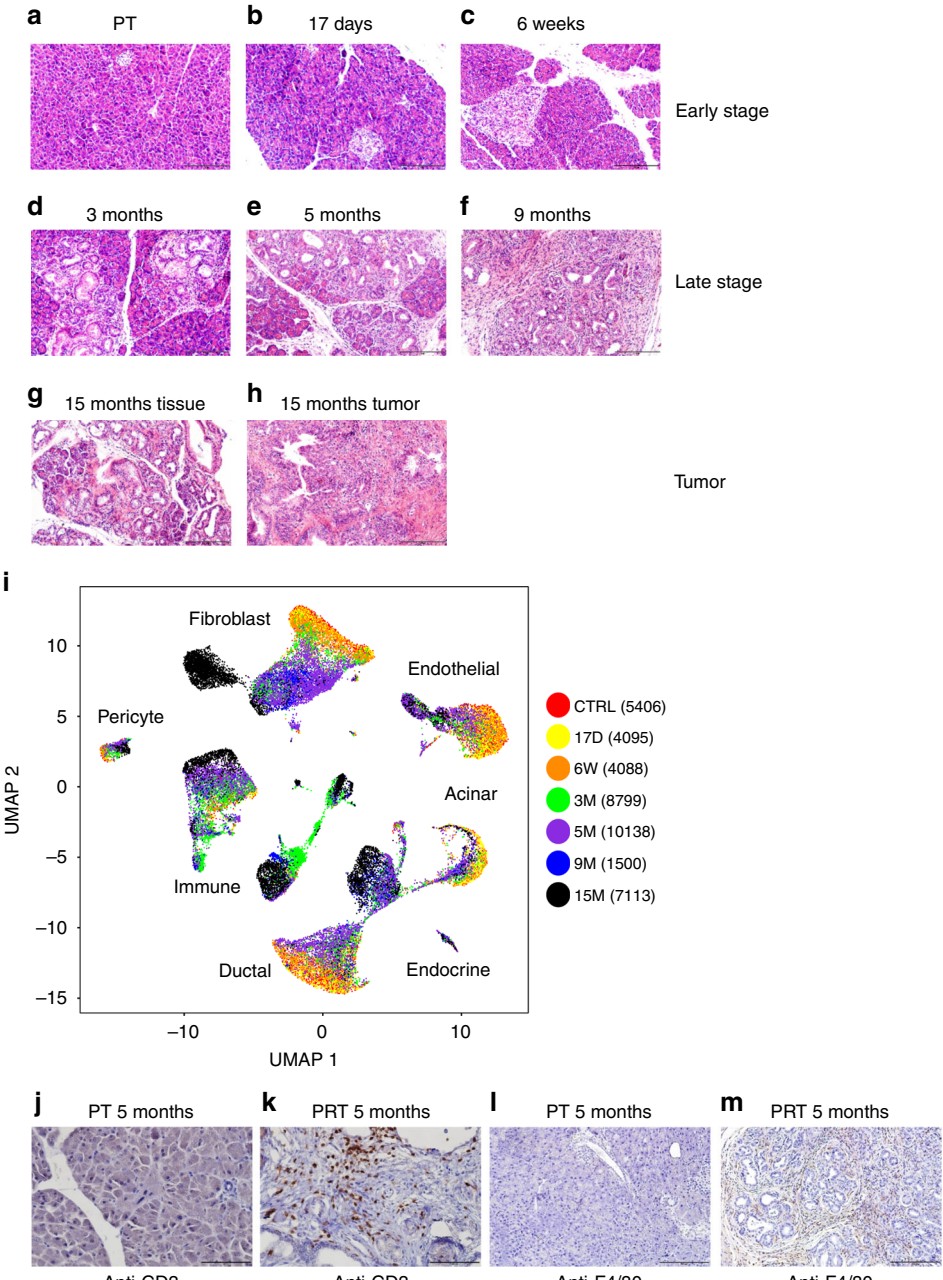

**Fig. 1 Single-cell RNA-seq experiment of pancreatic tissue, taken from *Ptf1a-CreER, LSL-Kras G12D, LSL-tdTomato* mice at different time points post-tamoxifen injection (PTI). a–h** The accumulation of duct-like structures and PanINs at different time points post-tamoxifen injection (PTI). Hematoxylin and Eosin (H&E) staining of histological sections of mice pancreas. The time point PTI is indicated above each panel. Based on the number of PanINs and duct-like structures (quantification in Supplementary Fig. 1a), we define early stage, late-stage and tumor sample. **g, h** Two sections from the same mice, **g** tumor adjacent tissue, and **h** tumor tissue. In early stage samples, lesions were rare. Scale bars 200 μm. **i** Pancreatic tissues from mice were dissected and single-cell RNA-seq experiments were performed. Uniform manifold approximation and projection (Umap) includes all cells from seven different time points PTI. The data were produced from nine mice, two mice from 3 months PTI, two mice from 5 months PTI and one mouse from each of the other time points. Time points are indicated on the right side of the panel, the number of cells is shown in parentheses. Cell types were determined based on the expression level of representative markers as indicated in Supplementary Fig. 1. **j, k** Immunostaining of pancreatic sections using anti-CD3 antibody. Both sections were taken from five months PTI mice, **j** control mouse *Ptf1a*-CreER, *LSL-tdTomato* (PT), **k** *Ptf1a-CreER, LSL-Kras G12D, LSL-tdTomato* (PRT) mouse. Scale bars 100 μm. **l, m** Immunostaining of pancreatic section using anti-F4/80 antibody. Both sections were taken from 5 months PTI mice, **l** control mouse *Ptf1a-CreER, LSL-tdTomato* (PT), **m** *Ptf1a-CreER, LSL-Kras G12D, LSL-tdTomato* (PRT) mouse. Scale bars 200 μm.

our experimental system supports the characterization and discrimination of authentic ductal and metaplastic *Krt19*-positive cells. We first analyzed the transcriptional changes in acinar or ductal cells and then the uniquely expressed genes in metaplastic cells.

The expression profiles of acinar and ductal cells consistently changed following tamoxifen injection (Supplementary Fig. 2). A comparison of the expression profiles of early and late acinar cells showed lower expression of genes that encode acinar secreted enzymes such as *Try4, Cpa1, Cpa2,* and *Cela2A* ($q$-value < 0.001)

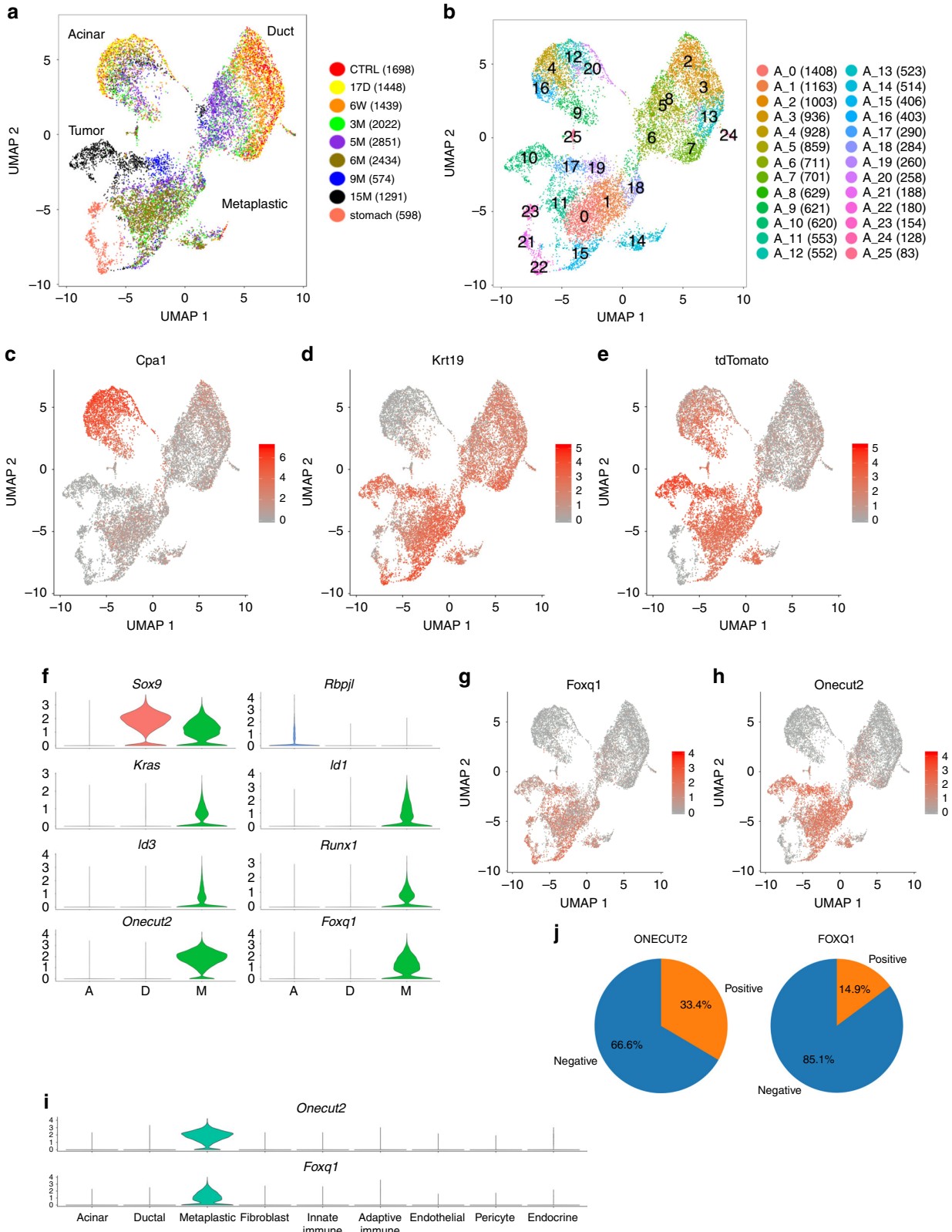

in late acinar cells (Supplementary Fig. 2j). Notably, starting 3 months PTI, acinar cells expressed high levels of *Reg* genes, including *Reg3b*, that inhibits the Stat3 signaling pathway[25]. In addition, late acinar cells express the *S100a* family genes, immune-related genes, such as *C4b* and *Cd74*, and the ductal marker *Krt19* (Supplementary Fig. 2j). When the expression profiles of late ductal cells and early ductal cells were compared,

we observed an increase in the expression of genes that promote inflammation and recruitment of immune cells (*C3*, *Tnf*, *Cxcl2*, *Fosl1*, *Il1rn*, and *Ltb*) (Supplementary Fig. 2j). These changes in expression programs reflect increased inflammation and stress from internal tissue injury.

Acinar metaplasia is considered a key event underlying PDAC development. Our system allows us to accurately discriminate

**Fig. 2 Metaplastic cells cluster separately compared to acinar cells and ductal cells. a** Analysis of the epithelial cells. UMAP, including ductal cells, *tdTomato*-positive cells, and stomach epithelial cells. Time points PTI and cell numbers are indicated on the right. Metaplastic and tumor cell clusters defined as *tdTomato*-positive *Cpa1*-negative cells. **b** The same UMAP as in (**a**), the number of each cluster is indicated. Cluster numbers and cell numbers in each cluster are shown on the right. **c–e** The expression levels of **c** *Cpa1* (acinar cells), **d** *Krt19* (ductal and metaplastic cells), **e** *tdTomato* (that mark cells that expressed *Ptf1a*). Based on the expression of these genes we define the clusters of acinar cells, ductal cells and acinar metaplastic cells. High expression level—red, low expression level—gray. **f** Violin plots showing the expression level of genes that are expressed by acinar, ductal, or metaplastic cells, A—include all acinar cluster cells (*Cpa1*-Positive), D—include all ductal cluster cells (*Krt19*-positive, *tdTomato* negative cells), M—include all metaplastic cluster cells (*tdTomato*—positive, *Cpa1*-negative). Gene's name is at the top of each chart. **g, h** Expression levels of *Onecut2* and *Foxq1*. The two transcription factors are expressed in metaplastic cells but not in ductal cells or acinar cells. **i** Violin plots showing the expression of *Onecut2* and *Foxq1* transcription factors in different cell types. The *x*-axis represents cell type based on the classification shown in Fig. 1. **j** Pei diagram showing the percentage of *KRT19*-positive cells that express *ONECUT2* and *FOXQ1* in a human PDAC sample.

---

between bona fide ductal cells versus metaplastic cells which express ductal markers. Metaplastic cells (*Cpa1*−, *tdTomato*+, *Krt19*+) comprised cell clusters distinct from clusters of acinar and ductal cells. They expressed high levels of *tdTomato*, thus originated from acinar cells, but low levels of typical acinar enzymes (Fig. 2e and Supplementary Fig. 2k). The metaplastic cells could only be detected at late time points starting 3 months PTI (Supplementary Fig. 2). To increase the number of metaplastic cells in our analysis, we performed an additional experiment in which *tdTomato*-positive cells were sorted from mouse pancreas 6 months PTI; of these, 2434 cells are included in the analysis presented in Fig. 2.

A total of 3020 differentially expressed (DE) genes were identified when metaplastic and acinar cells were compared, and a total of 1378 DE genes when metaplastic and ductal cells were compared (two-sided Wilcoxon test *q*- value < 0.01, Supplementary Data 2 and 3). We are specifically interested in the expression of transcription factors (TFs) that were highly expressed in metaplastic cells but showed very low or non-detectable expression in acinar and ductal cells and found five such TFs: *Id1*, *Id3*, *Runx1*, *Onecut2*, and *Foxq1*. The first three TFs were also expressed in non-epithelial cells in our data (Supplementary Fig. 3a), and *Id1* and *Id3* were previously reported to be expressed in PDAC samples[26]. *Onecut2* and *Foxq1* were specifically and highly expressed in more than 70% of metaplastic cells (Fig. 2f–h, Supplementary Data 4). *Onecut2* encodes a homeodomain protein containing a single CUT domain[27] that was recently reported to have a major role in prostate cancer[28]. *Foxq1*, a forkhead box (FOX) protein, is a TF that regulates the transcription of *Muc5ac*[29]. According to our data, *Onecut2* and *Foxq1* were not detected in any cell type in the pancreas other than acinar metaplastic cells (Fig. 2i, Supplementary Data 1).

In addition to the metaplastic population of cells mentioned above, we could also characterize early metaplastic cells that express *Cpa1*, *tdTomato*, and *Krt19* (denoted triple-positive). These cells could be divided into two clusters (Supplementary Fig. 3b). The transcription profile of the first cluster was similar to acinar cells (suggesting coincidental expression of *Krt19*), however, cells that were included in the second cluster (Supplementary Fig. 3b–c colored in red) had an intermediate transcription profile with partial overlap with the acinar transcription signature and partial overlap with the metaplastic transcription signature (Supplementary Fig. 3d). Comparing triple-positive cells and acinar cells (*Cpa1*+, *tdTomato*+, and *Krt19*−), the DE genes included *Sox4* and *Cdkn1a*, in addition to genes that regulate translation and genes that induce an immune response (logFC > 1, *q*-value < 0.05, Supplementary Data 5). Interestingly, the TFs *Onecut2* and *Foxq1* were expressed in the metaplastic cells but not in the triple-positive cells. These analyses suggest that *Onecut2* and *Foxq1* expression may be needed to regulate the transition from early metaplastic stage to late metaplastic stage (Supplementary Data 6). Consistent with this

possibility, we found a significant (*p* value = 0.00049) overlap between metaplastic expressed genes and Onecut2 chromatin binding sites based on published CHIP-seq data[30]. Overlapping gene sets may indicate that Onecut2 regulates cellular transformation and RAS signaling (Supplementary Fig. 3e).

To examine the relevance of these TFs to human PDAC, we performed scRNA-seq experiments of a sample from a human patient (Supplementary Fig. 4). A total of 5184 cells passed quality control. We identified immune cells, fibroblasts and seven *KRT19*+ clusters (Supplementary Fig. 4a–c) that had chromosomal copy number variation (CNV), based on the transcriptome, different from stromal clusters (Supplementary Fig. 4e). Each *KRT19*-positive cluster cell expresses a unique set of genes including chemokines and genes that encode proteins that regulate cell cycle progression (Supplementary Fig. 4f, Supplementary Data 7). *ONECUT2* and *FOXQ1* were expressed in 33.4% and 14.9% of *KRT19*+ cells, respectively (Fig. 2j). To examine the validity of these findings in a larger number of patients, we reanalyzed scRNA-seq data that was recently published by Peng et al.[24] (Supplementary Fig. 4g) and that includes data from normal pancreatic tissues and PDAC patients' tissues. Based on the analysis of *EPCAM*-positive cells, *ONECUT2* and *FOXQ1* were expressed in patients' cells but not in cells that were recovered from normal pancreata (Supplementary Fig. 4h, i). Using anti-Onecut2 antibody, we detected protein expression in tissue sections of mice and human PDAC samples (Supplementary Fig. 4k–m). In addition, high expression of *ONECUT2* was correlated with bad prognosis in tumor samples of PDAC patients (Supplementary Fig. 4j). Thus, *ONECUT2* and *FOXQ1* are expressed in the early stages of PDAC development and in tumors of humans and mice, and possibly play a role in the development of PDAC.

**Acinar metaplastic cells are heterogeneous**. Metaplastic cells comprise several distinct subpopulations of cells (Fig. 2a). Here, we found that the different clusters represent distinct metaplastic identities and states, and explored their expression profile. When we explored DE genes in metaplastic cells clusters, we found that *Gkn1*, *Gkn2*, and *Tff1* were among the most highly expressed genes in cells from a cluster denoted A_15 (Fig. 3a–d, Supplementary Data 4). These genes encode proteins that assist in protecting the stomach from acid-induced damage and are secreted by stomach pit (foveolar) cells. To explore the localization of *Gkn1*-positive cells found in cluster A_15, we used immunohistochemistry and surprisingly found that cells within PanIN lesions stained positive for Gkn1, while ductal structures composed of smaller cells not containing mucin were rarely stained (Fig. 3f–h, Supplementary Fig. 7h). These results suggest that *Gkn1*-positive cells in cluster A_15 are indeed localized to PanIN lesions. In contrast, anti-Krt19 stained cells localized to both ductal structures and PanIN lesions (Fig. 3i–k).

To explore the extent of similarity between metaplastic cells and stomach cells, we performed scRNA-seq of mouse stomach

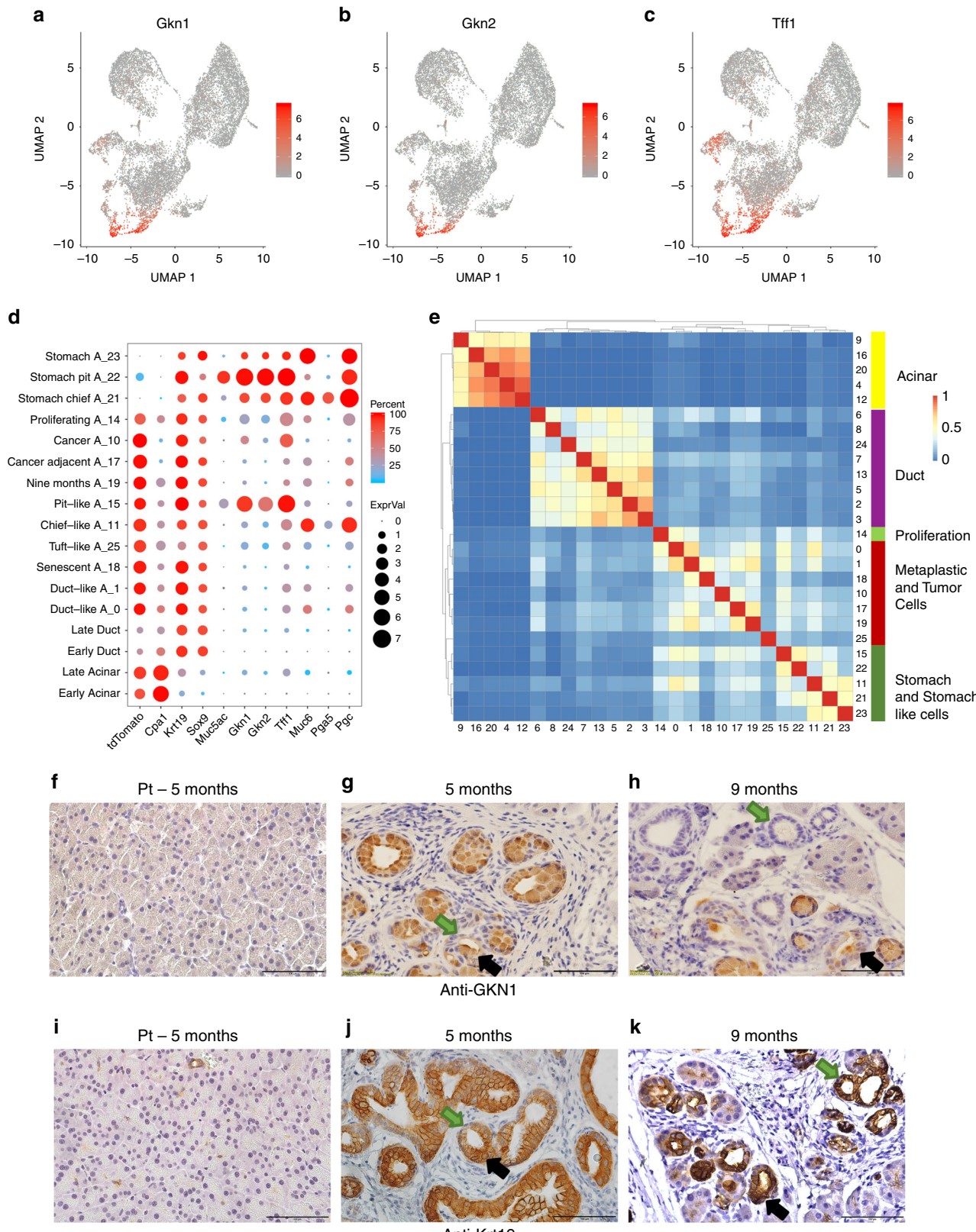

(Supplementary Fig. 5a–e). Clusters of stomach epithelial cells (expressing high levels of *Epcam*, *Krt19*, and *Sox9*) were included in the analysis of acinar, ductal and metaplastic cells (Fig. 2a, b, clusters A_21–23). Cells in cluster A_22 expressed *Gkn1*, *Gkn2*, and *Tff1*, and were therefore denoted as stomach pit cells (Fig. 3d).

We next examined if the shared markers between cells from different clusters indeed represent similarity in the transcription programs. We compared overlapping DE genes between all clusters in this analysis (see the "Methods" section) and found that cluster A_15 (originating from *tdTomato*-positive metaplastic cells from pancreatic samples) and cluster A_22 (originating

**Fig. 3 A subset of metaplastic cells express overlapping sets of genes with stomach pit cells, and are localized to PanINs. a–c** UMAP showing the expression level of highly expressed genes in cluster A_15 cells that we found to include Pit-like cells. In each chart, the expression level of one gene is indicated, **a** *Gkn1*, **b** *Gkn2*, and **c** *Tff1*. High expression level—red, low expression level —gray. Scale is on the right and relevant gene name on the top of each chart. **d** Dot plot showing the expression level of sets of genes (x-axis) in different subsets of cells (y-axis). Subsets of cells are based on the clusters in Fig. 2b early acinar/ductal cells (control, 17 days and 6 weeks PTI samples), late (3, 5, and 9 months PTI samples). The average expression level is shown by the size of the dot and the percentage of cells that express the relevant gene is indicated by the color. **e** Distances matrix of all clusters of acinar, ductal, metaplastic, and stomach epithelial cells shown in Fig. 2b. Cluster numbers are indicated on the bottom and on the right, clusters were ordered based on the hierarchical clustering analysis. The scale score of overlapping genes is shown in the red-blue bar. The cell types and cell lineages are indicated on the bar color on the right. Genes used for this analysis are based on comparing the expressed genes in the relevant cluster to all the cells shown in Fig. 1 (see "Methods"). **f–k** Histological sections of mice pancreas. **f, i** PT control mice, **g, j** five months PRT, **h, k** nine months PRT. **f–h** Staining with anti-Gkn1, **i–k** staining with anti-Krt19. Black arrows indicate representative cells that are large and therefore may contain mucins, green arrows indicate representative cells that are smaller and without mucins. Scale bars 100 μm.

---

from *Epcam*-positive normal stomach cells) had similar gene expression profiles (Fig. 3e and Supplementary Fig. 5f, Supplementary Data 8). The overlap of these profiles comprised genes such as *Cldn18*, *Muc5ac*, *Tff1* (Figs. 3c, 4k and Supplementary Fig. 5g, h), PanINs markers that are known to be secreted by stomach pit cells[31]. The overlap between the genes expressed in metaplastic pit-like cells and ductal cells was much smaller, although they include the known ductal markers *Krt19* and *Sox9* (Fig. 4a and Supplementary Fig. 5f). Importantly, more than 90% of the PanINs included at least one pit-like cell, while only 8.3% of the ductal structures included at least one pit-like cell (Supplementary Fig. 7i, Supplementary Data 9).

Further exploration of the subpopulations of metaplastic cells revealed *Muc6*-positive, *Krt19*-low cells in cluster A_11. Based on the correlation matrix, the transcription profile of these cells was close to that of cells in cluster A_21, which included stomach cells that we identified as chief cells, as they express *Pga5* and *Pgc*, which encodes pepsinogen (Fig. 4j and Supplementary Fig. 5i).

Thus, metaplastic cells express high levels of *tdTomato* and *Sox9*, but include several different subpopulations with similarity to ductal cells, metaplastic pit-like cells or metaplastic chief-like cells. Importantly, contrary to stomach cells, metaplastic cells did not express the *Sox2*, *Sox21*, *Gata4*, and *Bmp2* TFs, which play an important role in stomach differentiation[32] (Supplementary Fig. 5j). Therefore, these data do not support the notion that acinar cell metaplasia is mediated by stomach progenitor cells. In addition, none of the cell subpopulations expressed the stomach stem cell gene *Lgr5*; however, metaplastic cells expressed low levels of *Vil1* (Supplementary Fig. 5l), a marker of rare stem cells in the stomach[33].

We noticed an additional subpopulation of metaplastic cells that exhibited a distinct transcription profile in cluster A_18 (Fig. 2a, b and Supplementary Data 4). These cells had a senescent expression signature, based on the expression of *Cdkn2a*, P53 and P21 (Fig. 4a and Supplementary Fig. 5k), consistent with previous reports of senescent cells that localized to PanIN lesions[34]. In addition, cells in the acinar senescent cell cluster expressed genes that are involved in chemotaxis and gap junction, such as *Gjb4*, and pro-inflammatory cytokines, including *Il1a*, *Tnf*, *Tgfb1*, and *Il23a* (Supplementary Fig. 5k). Using anti-Tgfb1 we found that these cells were localized to the PanINs (Fig. 4c, f, g, i).

Cells in cluster A_25, which was relatively distant from the other *tdTomato*-positive cell clusters, expressed *Dclk1*, *Pou2f3*, and other genes, which encode known tuft cell markers (Fig. 4a, h and Supplementary Fig. 5k)[35,36]. Under normal conditions, tuft cells are not present in the pancreas, but upon Kras activation, acinar cells can undergo metaplasia to produce tuft cells[37–39], consistent with our finding that tuft-like cells were *tdTomato*-positive. Importantly, other cell types, including fibroblasts and ductal cells, expressed *Dclk1* (Supplementary Fig. 5m, n),

emphasizing the importance of using single-cell methods to profile the acinar metaplastic tuft cell transcriptome. In addition, our analysis revealed dozens of genes that are strictly expressed by cells in this cluster (Supplementary Data 4), including four TFs, *Spib*, *Hmx2*, *Pou2f3*, and *Hmx3* (Supplementary Fig. 5k). Tuft-like cells were localized to both ductal structures and PanIN lesions (Fig. 4c–e).

According to our analysis, some of the neuroendocrine cells expressed *tdTomato*. Reanalysis of neuroendocrine cells (Supplementary Fig. 6a–e) revealed that *tdTomato* was expressed in two different subclusters and that these cells originated from late time points PTI samples; we therefore suggest that these cells represent acinar-to-neuroendocrine metaplasia. We were able to validate that insulin-secreting cells were indeed localized to lesions (Fig. 4b).

The cells in the clusters mentioned above do not cycle, but some cells in the lesions do cycle (Fig. 4o, p). Indeed, cells in cluster A_14 had high cell cycle scores and expressed *Mki67* (Fig. 4a, l). We reanalyzed *Mki67*-positive cells and found 11 subclusters of dividing cells, nine of them containing *tdTomato*-positive cells (Supplementary Fig. 6f–j). Interestingly, the genes unique to the D_1 cluster include markers of pit cells, and cells in the D_2 cluster expressed *Muc6* and *Pgc* and may, therefore, represent proliferating stomach-like cells (Supplementary Fig. 6j, k). Thus, the subpopulation of chief-like metaplastic acinar cells and pit-like metaplastic acinar cells are proliferating. Other clusters of dividing cells were comprised of small numbers of cells precluding reliable cell lineage allocation.

Overall, our analysis of *tdTomato*-positive cells revealed that metaplastic cells are not a homogenous population but rather can be divided into distinct metaplastic lineages that infiltrate pancreatic lesions (Fig. 4). These *tdTomato*-positive cells included stomach pit-like cells (cluster A_15), stomach chief-like cells (cluster A_11), senescent cells (cluster A_18), tuft-like cells (cluster A_25), and neuroendocrine-like cells. Thus, we conclude that the previous designation, acinar-to-ductal metaplasia, does not fully encompass the extent of metaplasia.

Reanalysis of published human PDAC scRNA-seq data[24] revealed dominant expression of most of the metaplastic markers in separate clusters (Supplementary Fig. 7a, b). To further validate the expression of the metaplastic markers in human samples, we stained human PDAC and adjacent tumor sections and could detect TGFB1 and TFF1 in lesions and tumors (Supplementary Fig. 7c–f).

Based on our mice scRNA-seq data, we did not detect major changes in metaplastic cell composition at different time points PTI (3, 5, and 9 months), and stomach-like cells were the majority of the metaplastic cells (Supplementary Fig. 7g). Notably, the abundance of the metaplastic cell types in the ductal structures (ADM) was not equal to their abundance in PanIN lesions (Fig. 4c and Supplementary Fig. 7h, i,

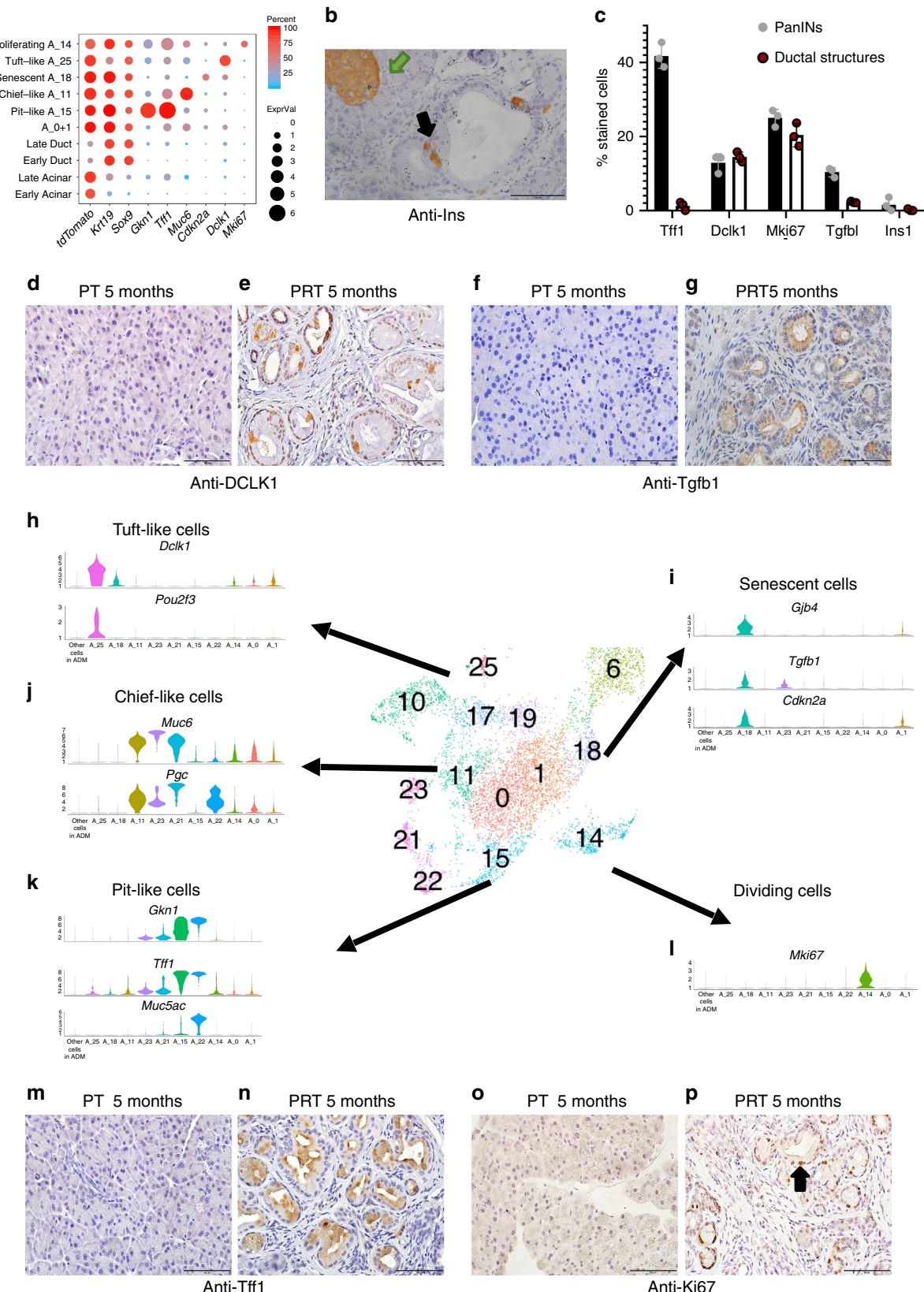

Supplementary Data 9). Stomach pit-like cells and senescent cells were enriched in PanIN lesions and most of these stained cells contained mucin (Fig. 4f, g, m, n), while tuft-like cells and proliferating cells could be detected in both ductal structures and PanINs (Fig. 4c and Supplementary Fig. 7h, i). Since it is not clear at this stage which of these metaplastic cells is the source of PDAC, and as PanINs are considered to develop from ductal structures, the different abundance of metaplastic cells in these lesions may shed light on the sequence of events that govern cellular transformation.

**Fig. 4 Acinar metaplastic cells are heterogeneous.** In the middle of the figure a snapshot of Fig. 2b is presented, showing *tdTomato*-positive and stomach cluster cells. **a** Dot plot of selected differentially expressed genes in early (control, 17 days and 6 weeks PTI samples) and late (3, 5, and 9 months PTI samples), acinar, ductal and metaplastic clusters of cells as indicated on the y-axis. Dot size represents the expression level and color represents the percentage of cells that express the indicated gene, in each cluster. **b** Histological sections of mouse pancreas 4 months PTI, immunostaining with anti-Ins1. A green arrow indicates an islet; a black arrow indicates a stained cell that is localized to PanIN. Scale bar 100 μm. **c** Quantities analysis of metaplastic cell-types and states in PanIN lesions and in ductal structures. Data collected from slices from three different mice for each antibody (n = 3), and is based on the counting of 24,808 cells, see Supplementary Data 9. The bars represent the mean values and ±SD. **d, e, h** Metaplastic tuft-like cells. Immunostaining using anti-Dclk1 antibody, **d** control, 5 months PT mouse, **e** 5 months PRT mouse, scale bars 100 μm. **h** Violin plot showing the expression of *Dclk1* and *Pou2f3*. **f, g, i** Senescent cells. Immunostaining using anti-Tgfb1, **f** control, 5 months PT mouse, **g** 5 months PRT mouse, scale bars 100 μm. **i** Violin plot showing the expression of *Gjb4*, *Tgfb1*, and *Cdkn2a*. **j** Metaplastic chief-like cells, violin plot showing the expression of *Muc6*, *Pgc*. **m, n, k** Metaplastic stomach pit-like cells. Immunostaining using anti-Tff1 antibody, **m** control, 5 months PT mouse, **n** 5 months PRT mouse, scale bars 100 μm. **k** Violin plot showing the expression of *Gkn1*, *Tff1*, and *Muc5ac*. **o, p, l** Metaplastic cycling cells. Immunostaining with anti-Mki67 antibody, **o** control, 5 months PT mouse, **p** 5 months PRT mouse, scale bars 100 μm. **l** Violin plot showing the expression of *Mki67*.

Our analysis presents the transcription profiles of each cell population. In addition, we showed that the different cell lineages can coexist in a single lesion based on co-staining, for example, with anti-Muc5ac (pit-like cells) and anti-Dclk1 (tuft-like cells) (Fig. 5a–c). This characterization is important for the understanding of low-grade lesion formation and progression to high grade lesions.

**Potential interactions between cells in the microenvironment.** To assess potential paracrine effects of metaplastic cells on their microenvironment, we assessed the expression of secreted effector molecules. We found that cells in several metaplastic clusters expressed Il1 family cytokines, including *Il1A*, which was expressed by pit-like cells and acinar senescent cells. We also found that senescent cells expressed *Il1rn*, which inhibits Il1 signaling. In addition, *Il18*, which was expressed by pit-like cells (Fig. 5d), can bind to Il18r that was expressed by senescent cells. This finding suggests that different metaplastic cells in PanIN lesions signal to each other. Consistent with this possibility, we found PanINs that included both *Gkn1*-positive cells (pit-like) and *Tgfb1*-positive cells (senescent cells), indicating that these can coexist in the same PanIN lesion (Fig. 5f–i). Importantly, using IPA databases (see "Methods"), we found that an Il18-induced program could be detected in senescent metaplastic cells including upregulation of *Tnf*, *Timp1*, and additional cytokines (Supplementary Fig. 8a), but not in other metaplastic cells that do not express *Il18r1*, suggesting that the expression of Il18 by metaplastic pit-like cells functionally affects the surrounding cells in a specific manner. CD4 T cells and NK cells also expressed *Il18r1* (Fig. 5e), and it is therefore possible that metaplastic cells can alter immune cell phenotype. Consistent with this possibility, Il18-induced program could be detected in CD4-positive T cells (Supplementary Fig. 8b). Another gene that encodes a secreted protein, *Fam3d* (*Oit1*), was expressed by metaplastic pit-like cells, while the relevant receptors Fpr1 and Fpr2[40] are expressed by neutrophils (Fig. 5e). It is worth noting that neutrophils and T cells could be detected close to metaplastic pit-like cells (Supplementary Fig. 8c–h), thus enabling effective ligand–receptor interactions.

*Cxcl17* was expressed by *tdTomato*-positive metaplastic cells and encodes a protein that was reported to bind to Gpr35[41]. *Gpr35* was expressed by neutrophils, macrophages and dendritic cells (DCs) in our data (Fig. 5d and Supplementary Fig. 10k). In addition, cells that infiltrate PanIN lesions express genes such as *Timp1*, which regulate extracellular matrix (Fig. 5d). Thus, metaplastic cells secrete molecules that signal to immune cells and shape the lesion microenvironment.

Overall, we delineate a heretofore unrecognized metaplastic heterogeneity in the pancreas and characterize potential signaling between metaplastic cells, and with other cells in the stroma.

**Characterization of *tdTomato*-positive tumor cells.** Next, we explored the transcriptome of cancer cells and their relation to the metaplastic cells. *TdTomato*-positive cells from nine months and 15 months PTI samples clustered separately from cells sampled at earlier time points (Fig. 2a). Cells in the 15 months PTI sample expressed genes that regulate the Wnt signaling pathway, including *Wnt4*, and genes that encode proteins that shape the extracellular matrix, such as *Mmp7* (Fig. 6b). *MMP7*, which was previously shown to be expressed in human PDAC samples and affects acinar cell apoptosis, metaplasia, and tumor progression[42,43], was also detected in the human PDAC sample analyzed in this study (Supplementary Fig. 4f). Additionally, the *Fkbp5* and *Msln* genes were strictly expressed in the 15 months *tdTomato* sample (Fig. 6b). Consistent with our data, *MSLN* was previously shown to be expressed in gastric cancer[44] and this gene product was used as a target for chimeric antigen receptor (CAR) T cells in clinical trials with PDAC patients[45,46]. Similarly, based on our data, highly expressed genes can be examined as additional targets for cellular immune responses.

The 15 months PTI *tdTomato*-positive cells were divided into two main clusters (cluster A_10 and cluster A_17). Based on CNV analysis (Fig. 6a), cells in cluster A_10 had increased copy number aberrations (CNV-positive; may include malignant cells), while cells in cluster A_17 had CNV similar to stromal and metaplastic cells (CNV-negative; may originate from tumor adjacent region). Next, we identified genes that are DE between CNV-positive clustered cells and CNV-negative clustered cells. CNV-negative cells expressed *Scnn1a* Fig. 6b, a subunit of a sodium channel that plays a role in the clearance of mucus. *SCNN1A* was also expressed by *KRT19*-positive cells in the human PDAC sample (Supplementary Fig. 4f). CNV-positive cells expressed *Krt20*, a marker of pancreatic cancer cells, and Aim2, which activates caspase 1 to process Il1 and Il18 precursors and regulates inflammasome activation (Fig. 6b). Interestingly, tumor cells expressed *Ltbr*, which encodes a receptor for *Lta* and *Ltb* that were expressed by late ductal cells and by several immune cell-types (Supplementary Fig. 2j, Fig. 5e), including CD4 T cells from the 15 months PTI sample. The same potential interaction was found by analysis of the human data, as *LTBR* was detected in cells from *KRT19*-positive cells, while *LTB* was expressed by human CD4 T cells (Supplementary Fig. 4d).

**Modeling the transition of acinar cells.** To explore the order of events along the metaplastic process toward cancer following the expression of constitutively active *Kras*, we performed trajectory analysis (Fig. 6c, h, see "Methods"). We expected to find continuous changes of acinar cells to metaplastic cells and then to tumor cells. Surprisingly, based on the analysis, acinar cells could proceed to two different parallel stages: (i) metaplastic

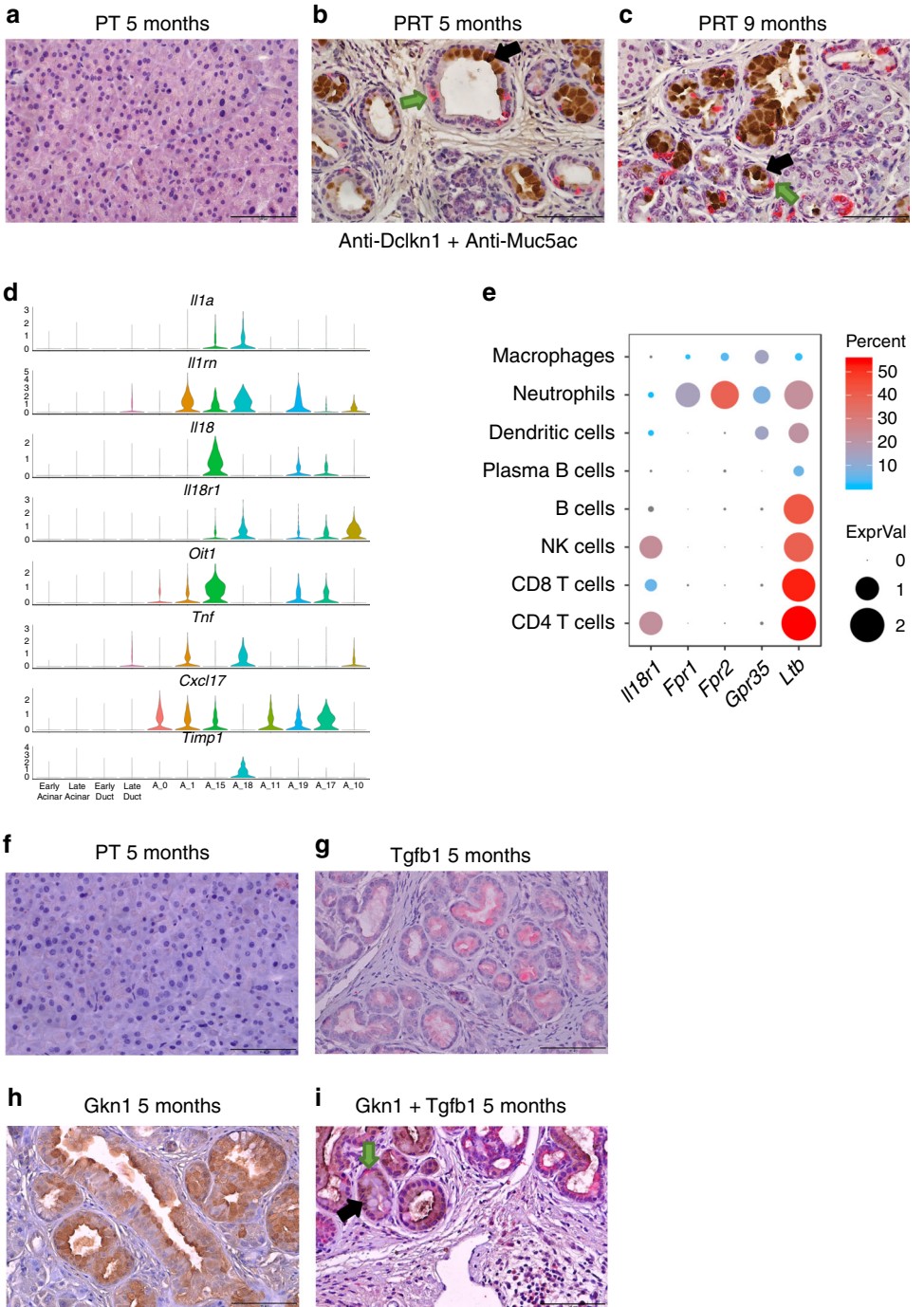

**Fig. 5 Potential metaplastic cell–cell interactions and co-localization of different metaplastic cell types to the same PanINs. a–c** Immunostaining of histological sections with anti-Dclk1 (red) and anti-Muc5ac (brown) antibodies. Arrows mark representative cells in the PanINs that stained positive with anti-Dclk1 (green arrow) or with anti-Muc5ac (black arrow). **a** Section taken from control, 5 months PT mouse, **b** section taken from 5 months PRT mouse, **c** section taken from 9 months PRT mouse. Scale bars 100 μm. **d** Violin plot showing the expression level (normalized expression) of a selected set of secreted molecules and receptors. On the x-axis: Early acinar/ductal cells (control, 17 days and 6 weeks PTI samples), late (all other time points), and clusters of metaplastic cells. **e** Dot plot showing the expression of a set of receptors (x-axis) across the different subset of immune cells (y-axis). Immune cells subtypes are determined based on data presented in Fig. 7. The average expression level is shown by the size of the dot and the percentage of cells is indicated by the color. **f, i** Immunostaining of histological sections of the pancreas, double stain with anti-Tgfb1 (red) and with anti-Gkn1 (brown), representative cell marked with arrows, anti-Tgfb1 (green arrow), and anti-Gkn1 (black arrow), **f** control, 5 months PT mouse, **i** 5 months PRT mouse. **g** 5 months PRT mouse section, stained with anti-Tgfb1, **h** 5 months PRT mouse section, stained with anti-Gkn1. **i**. Scale bars 100 μm.

stomach-like cells (state 2, Fig. 6e and Supplementary Fig. 9a) or (ii) tumor cells (state 3, Fig. 6d and Supplementary Fig. 9b). The possibility of parallel progression to the two states is also supported by pseudo time analysis (Supplementary Fig. 9c).

We next searched for DE gene signatures of each state with the aim of finding genes that, when expressed in late acinar cells, can predict cell fate, namely, whether this cell will acquire a state 2 gene signature (become a metaplastic stomach-like cell) or a state

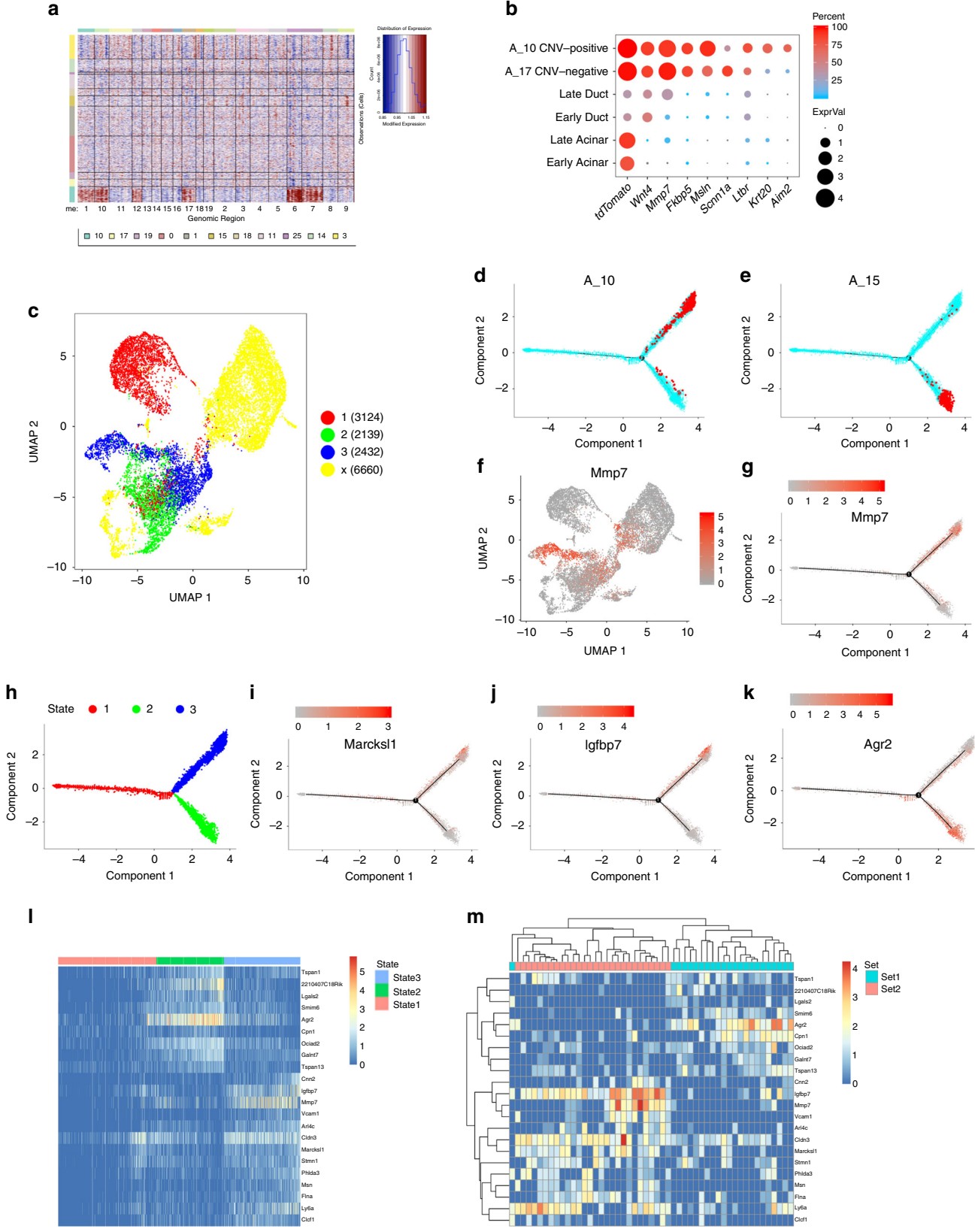

3 gene signature (tumor). We found a set of genes expressed in a subset of late time points PTI acinar cells (Fig. 6l), including *Mmp7*, *Marcksl1*, and *Igfbp7* (Fig. 6f, g, i, j). In addition, we found genes that were expressed in a different subset of late acinar cells at stage 1, including *Agr2* (Fig. 6k), which was reported to be

associated with PanIN and initial development of PDAC[47,48]. Importantly, the same set of genes that represent state 2 and state 3, clustered in an unsupervised way to two different populations of acinar cells (Fig. 6m), indicating two alternative programs at the very early stage, each expressed in different subsets of

**Fig. 6 Characterization of changes in metaplastic and tumor cells gene expression. a** CNV analysis of *tdTomato*-positive cells. On the *x*-axis mouse chromosome regions, on the *y*-axis clusters numbers. The copy number variation score is indicated on the right. **b** Dot plot showing the expression of a set of genes that are differentially expressed between normal acinar and ductal cells, and tumor cells (clusters A_17 and A_10). **c** Trajectory analysis of acinar ductal metaplastic and stomach epithelial cells colored according to states. The states are shown on the right, cells in yellow were not considered in the trajectory analysis. Numbers of cells are in parentheses. The figure shows that acinar cells can process to two different stages. **d**, **e** Trajectory charts showing: A_15 cluster cells (metaplastic pit-like cells) in red (**d**), and A_10 cluster cells (cancer cells) in red (**e**), all the other cells in the analysis in turquoise. **f** UMAP showing the expression level of *Mmp7*. Red—high expression; gray-low expression. **g** Trajectory chart showing the expression level of *Mmp7*. **h** Trajectory chart including the different stages. Coloring matches the colors in panel **c** and indicated above the panel. **i–k** Trajectory chart showing the expression level of **i** *Marcksl1*, **j** *Igfbp7*, **k** *Agr2*. *Marcksl1* and *Igfbp7* express in acinar cells and cells in state 3, *Agr2* expresses in acinar cells and cells in state 2. Red-high expression; gray-low expression. **l** Heatmap includes the differentially expressed genes between states 2 and 3. Differentially expressed genes were included if they expressed in cells from cluster A_9. Order of genes similar to the order in panel **m**. Cells from trajectory analysis included and ordered based on their state. **m** Heatmap showing the same differentially expressed genes as in panel **l**, genes and cells ordered by hierarchical clustering. Only cells from cluster A_9 were included.

cells and leading to different cell fates, stomach metaplastic or cancerous.

**Immune cell recruited simultaneously with PanINs formation.** Immune cells play a critical role in the development of PanIN lesions and PDAC[17,18]. We reanalyzed CD45-positive clusters and found that macrophages were the only immune cell type detected in our experiments at early time points and the largest population of immune cells at late time points (Fig. 7a, b and Supplementary Fig. 10a, b). Macrophages sampled at later times PTI, expressed genes that induce inflammation and chemokines such as *Cxcl1*, *Cxcl2*, and *Ccl8*, that attract monocytes, neutrophils, and additional macrophages[49] (Fig. 7c), and therefore amplify the recruitment of additional immune cells. Importantly, CD206 (encoded by *Mrc1*, M2 marker) expression, among other markers, was increased in tumor-associated macrophages (Fig. 7c and Supplementary Fig. 10c), showing that immune cells acquire a suppressive phenotype in this model.

Starting 3 months PTI and simultaneously with PanINs formation, there was an increase in all types of immune cells (Supplementary Fig. 10d–i). We found plasmacytoid DC (cluster I_29, high SiglecH levels), conventional DC (clusters I_5 and I_10) and a new subset cDC3 (high *Fscn1* and *Ccr7*) that was reported to infiltrate lung tumors in humans and mice[50], but has not been previously reported as PDAC infiltrating cells (Supplementary Fig. 10).

Cells in cluster I_24 showed an expression pattern that matched tumor infiltrating *SiglecF*-positive neutrophils that associate with cancer and may contribute to the immunosuppressive environment[50] (Fig. 7d and Supplementary Data 10).

Lymphocytes identified at late time points PTI and in the tumor sample, included B cells (Supplementary Fig. 10m), NK cells (Supplementary Fig. 10f), and T cells, that localized to PanINs lesions (Supplementary Fig. 10g–i, Supplementary Fig. 8c–f). *CD4* T cells in cluster I_16 include cells from all the late time points, and expressed both *Gata3* and *Foxp3*, the master TFs of Th2 cells and Tregs, respectively (Fig. 7e). These cells also expressed Pd-1 (*Pdcd1*) and *Ctla4*, while macrophages, DCs, neutrophils and plasma B cells, but not epithelial cells or other cells in the stroma, expressed PD-L1 (*Cd274*) (Supplementary Fig. 10j). Thus, *CD4*-positive T cells expressed genes that correlate with a suppressed phenotype, even at the pre-malignant stage, but *CD8*-positive cells did not (Fig. 7e, Supplementary Data 10).

Overall, immune cells start to accumulate together with PanIN formation, and signaling from PanIN epithelial cells and other cells in the stroma, affect their phenotype. The recruitment of immune cells is mediated by endothelial cells as we describe next.

**Endothelial cells express immune-attracting molecules.** When reanalyzing the endothelial cells, we noticed two states of expression profile and defined them as "quiescent", or "activated"—consistent with previous study[51] cells with enhanced expression of immune-attracting molecules and vascularization factors. Approximately, 94% of endothelial cells in a "quiescent" state originated from cells sampled at an early time point PTI, while 90% of endothelial cells in an active state originated from cells sampled at a late time point PTI (Fig. 7f, g and Supplementary Data 10). Endothelial cells in the active state, expressed (i) selectins (*Sele* and *Selp*) and adhesion molecules (*Icam*, *Vcam*, and *Madcam1*); (ii) chemokines (*Cxcl10*, *Cxcl2*, and *Cxcl1*); (iii) immune signaling regulators that control the inflammatory response (*Irf8*, *Tnfaip3*, and *Il6*); and (iv) genes that encode proteins with a structural role and a vascularization role, and *Hif1a* (Fig. 7h, Supplementary Data 10), a master regulator of the response to hypoxia[52]. In the early stage samples, *Il10rb* (but not *Il10ra*) was highly expressed in addition to the endothelial marker *Esm1* (Fig. 7h and Supplementary Data 10). Thus, even at the preinvasive stage, we could detect comprehensive changes in endothelial cells expression profiles.

Different cytokines and receptors are expressed on different subsets of endothelial cells. To determine which immune cells are attracted by endothelial cells, we matched receptor-ligand pairs. This analysis revealed a specific set of interactions; for example, "activated" early endothelial cells (cluster E_13, Fig. 7) expressed *Vcam1*, which encodes a receptor that binds Itga4 and Itgb1 (VLA-4)[53] that, according to our data, were expressed by dendritic and natural killer (NK) cells (Fig. 7i). Cells in cluster E_0 expressed *Cxcl1* and *Cxcl2*, which encode chemokines that bind Cxcr2[54]; therefore, based on our data, these cells may recruit neutrophils. Interestingly, the cells in cluster E_3 expressed genes that encode several receptors, chemokines, and cytokines, including *Csf3* that can interact with the gene product of *Csf3r*[55] that was expressed on neutrophils (Fig. 7i). In summary, the data from this analysis suggested that endothelial cells are activated along the course of disease progression and that subpopulations of activated endothelial cells attract specific immune cells.

**Fibroblasts along the development of lesions and tumors.** Cancer-associated fibroblasts (CAFs) produce a desmoplastic stroma that reduces the efficacy of chemotherapy and immunotherapy[56]. In addition, fibroblasts secrete growth factors that affect malignant cells, and cytokines and chemokines that can recruit immune cells and impose immunosuppression[57]. We analyzed the fibroblasts to profile the lesions and tumor-associated subpopulations at different time points PTI (Supplementary Fig. 11a, b).

When the gene-expression profiles of early PTI fibroblasts and late PTI fibroblasts were compared, we found that the expression of many genes that relate to collagen synthases and regulation was changed. These genes included *Timp1*, which inhibits the activity of collagenase[58]; *Lox*, *Plod2*, and *Bgn*, which are involved in

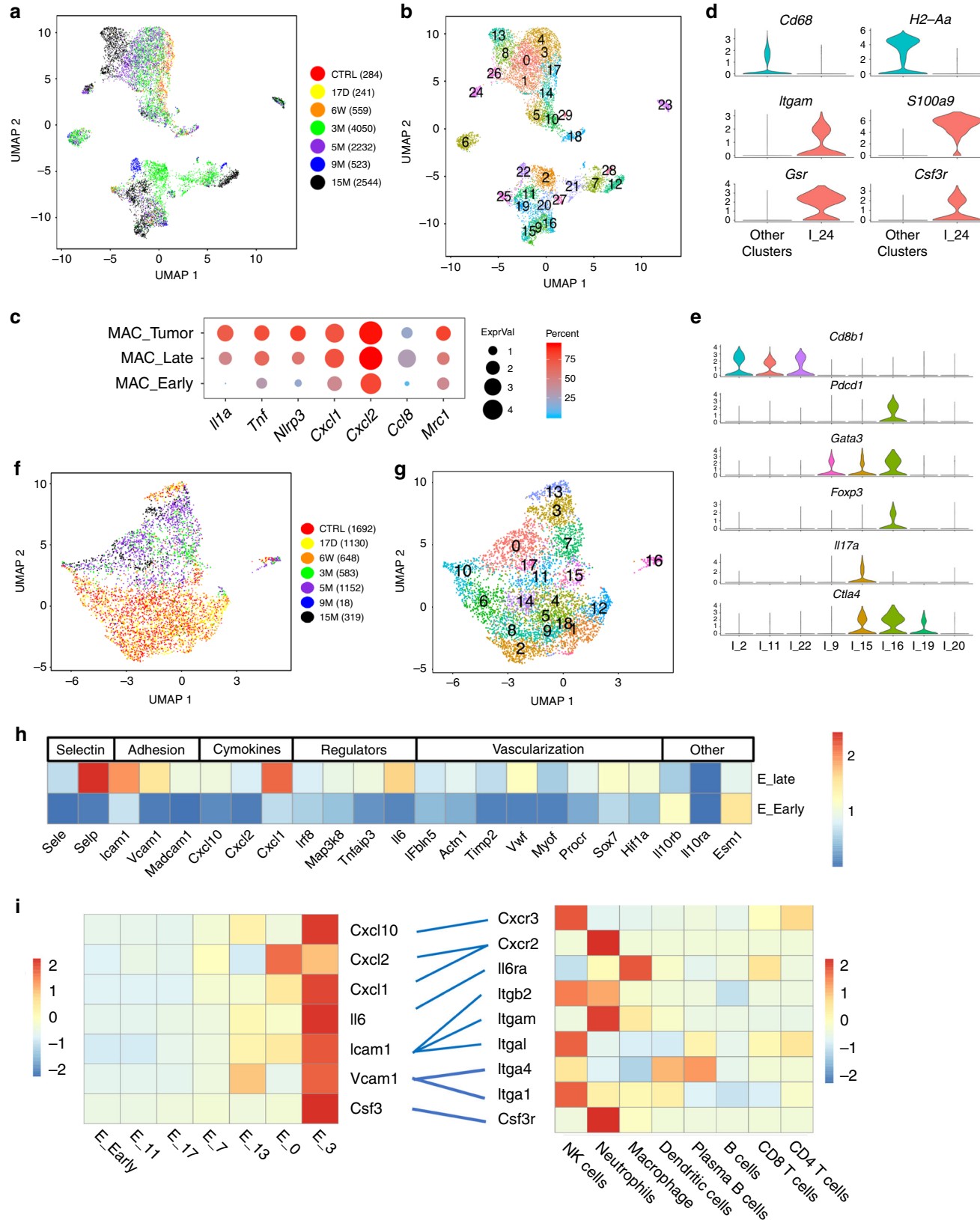

collagen fiber assembly[59–61]; and collagen production genes (Supplementary Fig. 11c–e). This finding is in accordance with one of the major phenomena of PDAC: formation of a desmoplastic microenvironment.

At early time points PTI, two clusters of fibroblasts were observed: *Igfbp5+* cells (cluster F_6) and *Il6+* cells (cluster F_5 and cluster F_1). These two subpopulations of fibroblasts expressed different sets of genes that encode cytokines and adhesion molecules (Supplementary Fig. 11f). Comparing early and late *Il6+* cells, we found that the latter, expressed higher levels of cytokines, genes that encode secreted proteins and enzymes, including *Ccl2*, *Ccl7*, and *Cxcl2* (Supplementary

**Fig. 7 Immune cell and endothelial cell heterogeneity. a** UMAP including *CD45*-positive cells. Time points post-tamoxifen injection (PTI) and cell numbers are presented on the right. **b** The same UMAP as in (**a**), the number of each cluster is indicated. **c** Dot plot of selected differentially expressed genes in macrophages; in early time point PTI, late time point PTI and tumor-associated macrophages. Dot size represents the scaled expression level and color represents the percentage of expressed cells. **d** Violin plot of cells in cluster I_24 (which we identify as neutrophils) and cells from all other CD45-positive clusters. Markers of macrophages and neutrophils are shown. Expression levels on the *y*-axis—normalized expression. **e** Violin plot showing the expression level of selected genes that are expressed in CD3-positive cells. Clusters I_2, I_11, and I_22 include CD8-positive T cells. Clusters I_9, I_15, I_16, I_19, I_20, include CD4-positive T cells. **f** UMAP includes CD31-positive cells. Time points post-tamoxifen injection and cell numbers are presented on the right. **g** The same UMAP as in (**f**), the number of each cluster is indicated. **h** The average expression level of selected differentially expressed genes of endothelial cells, between early time point PTI and late time point. Expression scale is indicated on the right and gene annotation on the top. **i** Potential interactions between endothelial cells and immune cells. Scaled gene expression of selected adhesion molecules, cytokines and chemokines and their matched receptors are shown. Blue lines show the potential interactions. Left heat map includes endothelial scaled expressed genes as indicated in endothelial cells clusters. Average of all the early time point PTI clusters or individual late time point PTI clusters as listed on the bottom. Right heat map includes expression levels of genes that encode matched receptors in immune cells as indicated on the bottom. Red - high expression, blue - low expression (see bars).

Fig. 11f). These genes were also highly expressed in *Il6+* fibroblasts recovered from the tumor sample (cluster F_8), and is consistent with the inflammatory role of inflammatory CAFs (iCAFs)[62]. At late time points PTI, we detected myofibroblasts (*Acta2*-positive cells) that expressed high levels of *Des* and *Igfbp5* compared to the IL6 + fibroblasts cells (Supplementary Fig. 11f), which is consistent with previous reports[63,64].

Three additional subpopulations of fibroblasts expressed distinct sets of genes (clusters F10-F12, Supplementary Fig. 11 and Supplementary Data 10): (i) proliferating fibroblasts (most cells expressed *Acta2*); (ii) fibroblasts that were recently discovered as *MHC-II* and expressed additional related genes such as *CD74* and *CD83*[23]; and (iii) 15 month's CAFs that expressed *Acta2*, *Tgfb1*, and *Cx3cl1*, and the members of the Wnt signaling pathway, such as *Wnt2*[65], most likely indicating the importance of these cells in signaling to other cell types in the lesion microenvironment (Supplementary Fig. 11f, g).

## Discussion

Several scRNA-seq studies in human PDAC have been published, that map cellular heterogeneity of samples from patients (see for example, Peng et al.[24]); however, scRNA-seq profiling of pre-malignant lesions is missing. Profiling early development of PDAC has great potential for improving early detection, which is challenging in pancreatic cancer and considered a leading cause of high mortality from this disease. Acinar metaplasia is an important early step in the development of PDAC, but our understanding of this process, the molecular identity of involved cells and their contribution to malignancy is partial. In our present study, using a genetically engineered mouse model, we have identified genes that were uniquely expressed in metaplastic cells, based on scRNA-seq of the pancreatic lesion. We found two TFs, *Onecut2* and *Foxq1*, that were expressed in late metaplastic epithelial cells originating from acinar cells, but not in early metaplastic cells or in any of the other cells in the pancreas (Fig. 2). *Onecut2* is a member of the one CUT homeodomain TF family. *Onecut1* (*Hnf6*), another member of this family, was suggested to be involved in pancreatic metaplasia lesion formation[66]. Notably, our data show that the expression patterns of *Onecut1* and *Onecut2* were opposite: *Onecut1* was expressed only in ductal cells (Supplementary Fig. 5o) but not in metaplastic or tumor cells, while *Onecut2* was expressed in metaplastic cells, mouse and human cancer cells (Fig. 2, Supplementary Fig. 4k–m) and is, therefore, more relevant to the development of PDAC that originates from acinar cells. *Onecut2* was recently shown to play a role in prostate cancer[28], but the role of this gene in PDAC has not been reported before. Based on our analysis, Onecut2 may regulate the expression of genes that control Kras downstream signaling. In addition, high expression of *ONECUT2* correlated

with bad prognosis, making *ONECUT2* a potential driver of early stage PDAC development.

*Foxq1*, a Forkhead-box TF, regulates the expression of *Muc5ac* in the stomach[29], however, in our data, the expression of *Foxq1* was not restricted to the metaplastic *Muc5ac* positive cells (Supplementary Fig. 5h), but rather was expressed in all the metaplastic clusters. *Foxq1* was reported to be expressed in several cancer types and may promote initiation, proliferation, invasiveness and metastasis[67], but its role in PDAC is poorly explored. Similarly to *Onecut2*, *Foxq1* was expressed in *Krt19+* human and mouse PDAC samples, emphasizing its potential importance. To prove causation of the involvement of these transcriptional factors, relevant mouse models not currently available should be used.

Although some genes, such as *Krt19*, *Sox9*, *Onecut2*, and *Foxq1*, were expressed in all metaplastic clusters cells, we found that metaplastic cells include several subpopulations with distinct transcription programs. Our approach of performing massive scRNA-seq at different time points allowed us to profile early and late metaplastic cells, accurately define their multiple identities, find metaplastic types that have not been previously profiled and are localized to PanINs, and explore their development over time.

Two metaplastic cell clusters included transcription signatures of stomach pit cells and stomach chief cells. This finding is interesting considering that both pancreas and stomach develop from the foregut precursors. Previous studies have shown that overexpression of the key pancreas TF *Ptf1a*, converts stomach and duodenum to pancreas in different model organisms[68,69], and knockout of *Ptf1a* induces stomach gene expression signatures in the pancreas[70]. It was proposed that reduction in *Ptf1a* is coupled with the induction of *Mecom*, a stomach-related TF that we found to be expressed in metaplastic cells (Supplementary Fig. 5p), resulting in the induction of KRAS-dependent transformation[70]. Our data suggest that even without direct manipulation of *Ptf1a*, constitutively active Kras can induce expression of the stomach gene signature. Notably, the presence of stomach-specific genes was shown in human PanINs[71], but metaplasia into specific types of gastric cells was not reported so far.

Based on the trajectory analysis that we performed (Fig. 6), acinar cells and early metaplastic cells showed a continuous change to one of two fates: tumorigenic or alternatively stomach metaplastic that mimics specific lineage (such as pit, chief). Thus, metaplastic acinar stomach cells do not necessarily have the potential to become malignant. Importantly, *Tff1*, *Gkn1*, and *Gkn2*, that are expressed in stomach pit-like cells, are considered stomach tumor suppressor genes[72] and knock out of *Tff1* in mice induces stomach cancer[73,74], supporting an antitumorigenic role of these genes. Previous work showed that the number of metaplastic tuft-like cells decreases along the transition from a preinvasive lesion to tumor[37]. Our data is consistent with these

findings, as we could not detect *Dclk1* and other markers of tuft cells in tumor samples and most of the metaplastic tuft-like cells did not contain mucins (Supplementary Fig. 7h).

These sets of evidence raise a question: which metaplastic cells or combinations of metaplastic cells in a lesion develop to malignant cells? The answer to this question is not clear yet, but finding and characterizing the acinar metaplastic cell subtypes is an essential step on the way. One possibility to consider is that PanIN-associated cells lose their metaplastic identity during their transition to malignant cells and therefore it is hard to detect metaplastic markers in malignant cells. Alternatively, specific cell states may be blocked from progression to KRAS-induced malignancy, rather than promoting it, in a manner similar to the senescence state. In this case, some metaplastic cells or cell types can be a "dead end" of the malignant progress.

It is of high importance to identify cells with tumorigenic potential at the early, preinvasive stage, and segregate them from morphologically similar cells that will not progress. We therefore searched for genes that are expressed in late acinar cells and in tumor cells but not in metaplastic stomach cells. Such genes may be potential markers for the beginning of the malignant process. *Marcksl1*, *Mmp7*, and *Igfbp7* were found to have the relevant expression profile and are therefore candidate markers for early detection (Fig. 6).

Stromal cells, including fibroblasts and immune cells, are critical for PDAC development. They signal to epithelial cells and in addition form an immunosuppressive and desmoplastic environment that protects malignant cells and enhances tumor growth. Based on the expression profiles, we found that endothelial cells and fibroblasts dramatically change together with the appearance of PanINs and that immune cells are recruited to the pancreas and the lesions.

Endothelial cells' phenotypic transition results in enhanced vascularization, which is probably needed in the damaged tissue and enhances the recruitment of other cell types, mainly immune cells. In addition to endothelial cells, fibroblasts, ductal cells and acinar metaplastic cells were found to express chemokines. We found potential interactions of receptor-ligands that may assist in recruiting neutrophils and macrophages, for example, Cxcr2–Cxcl1/2/5, Cxcl17–Gpr35, Csf3–Csfr3, and pairs of receptor-ligand that may be involved in immune signaling, including Ltbr-Ltb/a and Il6R-Il6. *Il18* was expressed by metaplastic pit-like cells, and the gene that encodes the Il18 receptor was expressed by CD4 T cells. We found these two cell types adjacent to each other, and Il18 downstream induced genes were expressed in CD4 T cells (Supplementary Fig. 8b). Importantly, myeloid cells and CD4 T cells expressed markers associated with a dysfunctional phenotype even in the preinvasive stage, suggesting that the suppressive environment precedes the development of cancer.

Overall, pancreatic metaplastic cells are a heterogeneous population that includes cells with high similarity to stomach cells, tuft cells, neuroendocrine cells, and that can have a senescence expression profile. We showed that: (i) the different subtypes are localized to ductal structures and PanINs; (ii) the premalignant lesions include more than one metaplastic cell type, at least in some cases; (iii) and based on expression profile, the metaplastic cells can interact with each other and with other cell types in the tissue. This insight will lead to the finding of the sequence of events that support the progression of PDAC and the origin of the malignant cells. Comparing the effect of mutant Kras in different tissues may suggest new common downstream targets for treatment.

## Methods

**Human samples.** Human PDAC resection specimens were obtained from the Hebrew University-Hadassah Medical Center (Jerusalem, Israel). The studies were conducted in accordance with ethical guidelines (0327-17HMO Declaration of Helsinki). The study was approved by the Hadassah Medical Center committee for human experiments (Helsinki Committee). All patients provided written informed consent to analyze tumor tissue following resection without impeding pathological analysis. Tumor tissues were taken from patients with biopsy-proven PDAC. There was no selection bias involved.

**Mice.** The joint ethics committee (Institutional Animal Care and Use Committee) of the Hebrew University (Jerusalem, Israel) and Hadassah Medical Center (Jerusalem, Israel) approved the study protocol for animal welfare MD-18-15417-5 "Tissue dynamics in pancreatic cancer in mice", and we have complied with all relevant ethical regulations for animal testing and research.

The following mouse strains were purchased from The Jackson Laboratory: stock #007908[75], stock #019378[76], and stock #00817[77]. These mice were crossed to create: PRT (*Kras+/LSL-G12D; Ptf1a-Cre^ER; Rosa26^LSL-tdTomato*) mice and PT (*Ptf1a-Cre^ER; Rosa26^LSL-tdTomato*) mice. For single-cell RNA-seq of the stomach, we used a 9-week-old female, C57BL/6JOlaHsd that was purchased from Envigo. Experimental and control mice were co-housed. All mice were housed in SPF conditions.

The Hebrew University is an Association for Assessment and Accreditation of Laboratory Animal Care International–accredited institute.

Tamoxifen dissolved in corn oil and injected to adult mice (6–8 weeks females and males), subcutaneously on days 0 and 2 at a dose of 400 mg/kg. Following tamoxifen injection, mice were examined twice a week. Tumors are internal and could not be measured; therefore, mice were euthanized if abnormal clinical signs were observed according to the ethical protocol. Euthanasia was performed at different time points PTI as indicated in Fig. 1, using isoflurane and cervical dislocation.

**Tissue dissociation cell isolation and sorting.** After euthanasia, the mouse pancreas was removed and washed in HBSS × 1 (Biological Industries) and cut into 1–2 mm pieces. The pieces were suspended in 0.02% trypsin C-EDTA 0.05% (Biological Industries) for 10 min in 37 °C with agitation and washed with 10% FCS/DMEM. For the next dissociation step, the cells were washed with HBSS × 1 containing 1 mg/ml collagenase P (Roche), 0.2 mg/ml bovine serum albumin (BSA) (Sigma Aldrich), and 0.1 mg/ml trypsin inhibitor (Roche). Samples were incubated for 20–30 min at 37 °C with agitation, then pipetted up and down 10 times and returned to 37 °C. After an additional 10 min, samples were pipetted up and down and passed through a 70 mm nylon mesh (Corning #431751) and washed twice with HBSS×1 containing 4% BSA and 0.1 mg/ml DNase I (Roche). The samples were divided into 2 equal volumes, one sample was centrifuged and washed 3 times at 60*g* to isolate the large cells containing the acinar cells. The second sample was centrifuged and washed 3 times at 300*g* to collect all cells. When large numbers of red blood cells were observed within the second sample, the cells were treated with red blood cell lysing buffer (Sigma Aldrich). If clumps were observed, we treated the sample with trypsin again, as described above. Viability was detected using trypan blue under the microscope and if the viability was lower than 80%, live cells were isolated using the MACS dead cells removal kit (Miltenyi Biotech #130-090-101). The two samples were then combined at a cell ratio of 30% cells from the 60*g* samples and 70% cells from the 300*g* samples. One sample was sorted for *tdTomato* using BD FACSAria III instrument and collected in PBS.

Isolation of mouse primary gastric epithelial cells was performed from a 9-week-old C57BL/6 mouse. The stomach was opened along the greater curvature and washed five times with HBSS × 1 without calcium and magnesium, the tissue was cut into 3 mm pieces and incubated for 20 min in HBSS×1 containing 0.2 mg/ml collagenase P, 20 mM HEPES (Invitrogen #), 0.2% BSA and 20 mM MEM. The cells were washed with HBSS × 1 containing 4% BSA and incubated in HBSS X1 containing 0.2% BSA and 0.5 mg/ml Dispase II (Roche #D4693) at 37 °C with agitation, then pipetted up and down 10 times and passed through a 70 mm nylon mesh. After two more washes with HBSS × 1 containing 4% BSA and 0.1 mg/ml DNase I, the cells were counted.

Tumor specimens from human patients were cut into 1–2 mm pieces and the cells were isolated as above but with longer cell dissociation duration (about 1 h). Before loading the samples to the 10× genomics machine, cells were washed and counted. We loaded 6000–9000 cells for each sample.

We used Chromium Single Cell 3′ Reagent Kit V2 and V3 (10× Genomics) and prepare libraries for sequencing following manufacturer instructions. Sequencing on Illumina HiSeq 2500 and Nextseq500 platforms was performed as follows 26 bp (Read1) and 58 bp (Read2).

**Immunohistochemistry staining.** Tissues were fixed in 4% formalin, embedded in paraffin and sections were cut and mounted onto slides. Staining of mice tissue was performed with the following antibodies: anti-Muc5ac (Abcam #ab3649 1:100), anti-Cytokeratin19 (Abcam #ab52625 1:200), anti-ki67 (Abcam #15580 1:100), anti-Gkn1 (Thermofisher #PA547913 1:100), anti-Tgfbeta (Abcam #ab92486 1:100), anti-CD3 (Bio-Rad #MCA1477 1:250), anti-Ly6g (BD Pharmingen 551459 1:200), anti-CD3 (Abcam #ab21703 100ul), anti-Onecut2 (Abcam #ab28466 1:50), anti-F4/80 (Biorad #MCA497RT 1:200), anti-Tff1 (Abcam #ab190942 1:200),

anti-Dcamkl1 (Abcam #ab37994 1:50), and anti-insulin (Dako, A0564, GP, 1:10) overnight at 4 °C.

ImmPACT DAB peroxidase substrate (Vector Laboratories #) and ImmPACT Vector red alkaline phosphatase (Vector Laboratories #VE-SK-5150) were used as substrates. Hematoxylin (Vector Laboratories #H-3404) was used as a counterstain.

We used N-Histofine kit, mousestain kit 414321F when staining with anti-Muc5ac (mouse antibody).

**Cell counting and statistics.** Cells stained for Tff1, Dclk1, Mki67, Tgfb1, and Ins1. Positive-stained cells and unstained cells were counted, two images of each slide, from three mice for each antibody. The cells were counted and classified by lesion types: ductal structures of cells that do not contain mucins (ADM) and PanINs that include cells which contain mucins. Figure 4c, Supplementary Fig. 7h–i were created based on the counting of more than 42,000 cells that are summarized in Supplementary Data 9. The statistical analyses were performed using GraphPad Prism version 8.0.0 for Windows, GraphPad Software, San Diego, CA, USA, www.graphpad.com.

**Data preprocessing.** Sequencing output reads were converted to FASTQ files using bcl2fastq (v1.8.3) and aligned to the mm10 reference genome, supplemented with the coding sequence of the *tdTomato* gene, using the Cell Ranger pipeline (v3.0.2 10x Genomics). Filtered raw counts data was imported into Seurat R package (v2.3.4)[78] for further processing and analysis. Raw transcript counts of gene–cell matrices were filtered to remove cells with fewer than 200 transcripts; cells with total UMI counts lower than 1000; and cells with more than 5% (V2 chemistry, 10× genomics kit) or 10% mitochondrial genes (V3 chemistry, 10× genomics kit). In addition, genes expressed in less than three cells were removed from the analysis. About 3% of the cells were suspected as doublets, based on the expression of mixed cell type markers, these cells were also removed from the analysis. The UMI counts matrices were then log normalized and scaled with Seurat's NormalizeData and ScaleData functions.

**Data analysis.** The Seurat package was applied to identify major cell types using dimension reduction followed by clustering of cell groups. Genes with the highest variance were used to perform linear dimensional reduction (principal component analysis), and the number of principal components used in downstream analyses was chosen considering Seurat's PCHeatmap and Elbowplot. Seurat's unsupervised graph-based clustering was performed on the projected PC space[78].

**Dimension reduction and clustering analysis.** We used Seurat to perform graph-based unsupervised clustering, uniform manifold approximation and (UMAP) (McInnes, L., Healy, J., Saul, N. & Großberger, L. UMAP: uniform manifold approximation and projection. J. Open Source Softw. 3, 861 (2018)) and t-stochastic neighbor embedding (tSNE), for data visualization in two-dimensional space.

*P* values presented in Supplementary Data were calculated using one-vs.-rest Seurat's FindAllMarkers function configured with Wilcoxon signed-rank statistical test (two-sided test), min.pct = 0.15 and thresh.use = 0.15. After cell type identification, subpopulations were reanalyzed separately by repeating the procedure described above. DE genes were calculated with the Wilcoxon test to explore subclusters transcriptional profile and Seurat's FindMarkers function was used, with default settings, to compare DE genes between two clusters. All heat maps were drawn using pheatmap R package (v1.0.12) with default settings. A dot plot function was self-implemented in R using package ggplot2 (v3.3.0) and Seurat's AverageExpression function. The Venn diagram plot was created using (Larsson J (2020). *eulerr: Area-Proportional Euler and Venn Diagrams with Ellipses.* R package version 6.1.0, https://cran.r-project.org/package=eulerr).

"Triple-positive" cells were defined as all cells that have positive UMI counts for tdTomato, Krt19, and Cpa1. Reanalysis of the triple-positive cells yielded two different clusters as shown in Supplementary Fig. 3. Using Seurat AddModuleScore, we calculated the signature for acinar, ductal, metaplastic and triple-positive cells (only the early metaplastic). The 30 top DE genes for each cell-type were considered for calculation of the signature scores. The values of each signature were normalized using min max normalization and using pheatmap for showing the relative score.

**Geneset analysis.** Identifying the cell cycle stage of cells was done using Seurat's CellCycleScoring function and a gene-set of known cell cycle genes[79]. To identify the inflammatory signature, we used Seurat's AddModuleScore and score cell expressing an inflammation-associated gene-set (GO-0002437).

IL18 canonical pathways were analyzed using ingenuity pathway analysis (QIAGEN Inc., https://digitalinsights.qiagen.com/products-overview/discovery-insights-portfolio/analysis-and-visualization/qiagen-ipa/).

The input data for Supplementary Fig. 8a, were expressed genes in cluster A_18 compared to expressed genes at early time point acinar clusters cells with two-sided Wilcoxon test, logFC>0.5. The input data for Supplementary Fig. 8b, were expressed genes in CD4 T cells (I_15, I_16, and I_9) that were compared to the rest of the T cells clusters with a false-discovery rate < 0.05.

**CNV prediction.** CNV was inferred from RNA sequencing data using inferCNV R package (v1.0.4) (https://github.com/broadinstitute/inferCNV). In human tumor samples, K19-expressing cells were considered as tumor and compared to fibroblasts, which were presumed to be nonmalignant. In mice samples, no tumor/normal population was defined.

**Batch effects.** To explore batch effects between different samples, we analyzed two samples taken from 5 months PTI single-cell data sets and two samples taken from three months PTI single-cell data sets. We sampled the same number of cells from each repeat (Supplementary Fig. 1j, k).

**Overlap genes between different cell types.** To measure the similarity between cells transcription profile from different clusters (Fig. 3e), Shorenson similarity coefficient was used taking DE genes from each of the clusters in Fig. 2b vs. all pancreatic cells in the analysis. To visualize the overlap between genes in cluster A_15 vs. A_22 and ductal cells, Venn diagram was used (Supplementary Fig. 5f).

**Overlap of ONECUT2-binding sites with metaplastic gene lists.** ONECUT2 ChIP-seq binding data were obtained from previously published data[30]. Peaks were mapped to gene TSS using ChIPseeker R package (v3.11)[80] and genes located in 100 kb range from significant Onecut2-binding peak were considered as "Onecut2-bound" genes. This list of genes was overlapped with genes highly expressed or repressed in metaplastic cells (using a two-sided wilcoxon test: metaplastic clusters vs. the rest with abs(logFC) > 0.2). Overlap enrichment significance was evaluated using a Hypergeometric test (using SciPy python package (v1.4.1)[81]).

**Trajectory analysis.** Trajectory analysis was performed using a monocle R package (v2.9.0)[82] on cells with *tdTomato*+, including clusters: A_0, A_1, A_4, A_9, A_10, A_11, A_12, A_15, A_16, A_17, A_18, A_19, and A_20. Ordering genes were assigned according to the top 30 expressed genes from each cell cluster. Data were processed by DDRTree algorithm for reducing the dimension[83] and pseudotime was calculated for each cell. DE genes between cells in states 2 and 3 were calculated and assigned into two sets using Seurat's FindMarkers function. Genes with average logFC higher than 0.4 and less than 50% expression in other states were included (pct.2 < 0.5).

For each geneset a score was calculated, using Seurat's AddModuleScore function. Late acinar cells (cluster A_9 with cells from samples ≤ 6 M PTI PRT mice ), having scores higher than 0.05 for one of the sets, were examined. For Fig. 6h, i, cells that expressed less than 5 genes and genes that were expressed in less than 10 cells were filtered out.

**Statistics and reproducibility.** *P* value for Kaplan–Meier plot (Supplementary Fig. 4j) was derived using the Log-rank test, test-statistic is chi-squared under the null hypothesis.

To produce Fig. 4c, we counted a total of 24,808 cells from 3 different mice (*n* = 3) for each antibody. To produce Supplementary Fig. 7h, we counted a total of 39,241 cells from three different mice (*n* = 3) for each antibody. To produce Supplementary Fig. 7i, we counted a total of 837 lesions from three different mice (*n* = 3) for each antibody.

Duplicate single-cell RNA-seq samples of pancreatic tissue at three months PTI and 5 months PTI showed high similarity as shown in Supplementary Fig. 1j, k.

Following immunostaining with each of the antibodies used in the study, we took photos from at least three different fields from each mouse. We have used the following numbers of mice for each panel:

Figure 1a: *n* = 7, Fig. 1b: *n* = 2, Fig. 1c: *n* = 4, Fig. 1d: *n* = 4, Fig. 1e: *n* = 5, Fig. 1f: *n* = 1, Fig. 1g, h: *n* = 1, Fig. 1j: *n* = 3, Fig. 1k: *n* = 3, Fig. 1l: *n* = 2, Fig. 1m: *n* = 3, Fig. 3f: *n* = 3, Fig. 3g: *n* = 5, Fig. 3h: *n* = 1, Fig. 3i: *n* = 3, Fig. 3j: *n* = 3, Fig. 3k: *n* = 1, Fig. 5a: *n* = 1, Fig. 5b: *n* = 1, and Fig. 5c: *n* = 1, Fig. 5f: *n* = 3, Fig. 5g: *n* = 3, Fig. 5h: *n* = 3, Fig. 5i: *n* = 3, Supplementary Fig. 4k: *n* = 1, Supplementary Fig. 4l: *n* = 1, Supplementary Fig. 4m: *n* = 4, Supplementary Fig. 7c, d: *n* = 4, Supplementary Fig. 7e, f: *n* = 2, Supplementary Fig. 8c, d: *n* = 3, Supplementary Figs. 8e and 1f: *n* = 3, Supplementary Fig. 8g, h: *n* = 3.

We present photos of the histology section from different mice in the Data Source file. In addition, we present in the Data Source file, the FACS gating strategy of the sorting of the tdTomato-positive cells for the single-cell sample shown in Fig. 2.

**Reporting summary.** Further information on research design is available in the Nature Research Reporting Summary linked to this article.

## Data availability

The data discussed in this publication have been deposited in NCBI's Gene Expression Omnibus[81] and are accessible through GEO Series accession number GSE141017. Any additional raw data is available on request from the authors. Source data are provided with this paper.

## Code availability

No custom code or mathematical algorithm is used, name and version of the software are included in the method.

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

## Acknowledgements

We would like to thank Dr. Yuval Dor, Dr. Eli Pikarski, Dr. Zvika Granot and Dr. Ittai Ben-Porath for fruitful discussions and advice. We thank all members of the Parnas lab, Dr. Gillian Kay for English editing, Dr. Sharona Tornovsky-Babeay, Dr. Ilan Stein, Nofar Rozenberg and Yelena Piontek and Housam Husseini for expertize and support with IHC. Dr. Zakhariya Manevitch, for the tutorial and support in the microscopy unit, Dr. Eleonora Medvedev cell sorting and Shaul Horwitz mice holding. This project has received funding from the European Research Council (ERC) under the European Union's Horizon 2020 Research and Innovation program (grant agreement No. 758735 O.P.), Israel Science Foundation—Broad Institute Joint Program (grant No. 2621/18 O.P.), Israel Science Foundation grant (No. 526/18 O.P.), the Alex U. Soyka Program and grants from the Israel Cancer Research Fund (ICRF), Project Grant (G.Z.).

## Author contributions

O.P. conceived and designed the study and mentor the performance of the work and experiments. O.P., Y.S., O.Y.L. wrote the paper, with input from all authors. Y.S. lead the computational effort and perform data analysis, with help from R.Z.G. (CNV, human data and general support), Y.N. and S.E. (gene annotation). O.Y.L. perform the immunohistochemistry experiments and stomach scRNA-seq. D.K.G. conduct the single-cell dissociation protocol and perform the experiments together with O.Y.L. and R.K. and with help from L.X., L.P., and O.A. A.N. and I.S. prepared scRNA-seq libraries. K.A. assist in pathology analysis. G.Z. obtained human samples and mentor some of the experimental work.

## Competing Interests

The authors declare no competing interests.
