## [Peer Review File · Nature Communications]

Reviewers' comments:

Reviewer #1 (Remarks to the Author):

Summary

The manuscript entitled 'Single cell transcriptomes of pancreatic pre-invasive lesions and pancreatic cancer reveal the full heterogeneity of acinar metaplastic cells' by Schlesinger et al. reveals interesting observations regarding the plasticity of acinar cells during the early stages of Kras-driven transformation in pancreatic cancer. Although a similar analysis has recently been performed by the Wu lab (Peng et al., 2019 Cell Research), the approach taken by the authors of this manuscript led to interesting conclusions that stand distinct. In particular, the previous study focused primarily on building an atlas using human pancreas ductal adenocarcinoma (PDAC) tissue, while the work by Schlesinger and colleagues used tdTomato lineage labeling of acinar cells in a mouse model. Additionally, the authors looked to establish new routes of plasticity within acinar cells, while previous work focused on ductal cell sub-populations. This important detail sheds light on how acinar cells might take different routes to transdifferentiate into ductal-like cells, and therefore represents a significant step forward in our understanding of acinar-to-ductal metaplasia. In summary, the novel approach (fluorescent tagging of acinar cells) is one that can only be performed in mouse models and has gleaned novel insights into cellular plasticity of the pancreas. However, I have the following major and minor concerns:

Major Concerns:

1. The 'metaplastic cell' population, defined by (Cpa1-, tdTomato+, Krt19+) will specifically select for late ADM cells that have completely lost their acinar cell identity (Cpa1-), but what about triple positive (Cpa1+, tdTomato+, Krt19+) cells that would mark acinar cells as they undergo initial ADM. This critical difference might account for why the authors are finding such distinct differences in the cell populations. For a more complete analysis, the authors should examine and discuss earlier ADM cell populations using the above mentioned triple positive cells.
2. The authors have defined heterogeneity via scRNA-seq, but the data lack depth. In particular, data discussed (i) on page 6 in regard to Figure 2 and (ii) on page 10 in regard to Figure 5 are lacking. Can the authors directly show any functional importance of the short list of transcription factors (Figure 2) and secreted factors (Figure 5) discussed? Functional data, in vitro and/or in vivo, would strengthen the authors' conclusions in these key figures. For example, expression of ligand and receptor pairs (Figure 5) does not necessarily mean that those are the factors truly responsible for immune cell recruitment. Additionally, this data could be strengthened by correlative studies in tissue.
3. The discussion is critically lacking in a comparison to demonstrate how this current work is distinct from the work previously published by the Wu lab (Peng et al., 2019 Cell Research), which is relatively easy to do.

Minor Concerns:

1. More details should be provided on page 6-7 to explain the rationale for narrowing down to the 5 transcription factors discussed in more detail (Figure 2F-H). Are previously known transcription factors also identified?
2. Supplemental Figure 1L should be expanded to show tdTomato tracking over the different time points of isolation, to potentially show an evolution of the acinar-to-ductal metaplasia event. This might also capture acinar reprogramming events that are occurring.
3. Why are more conventional markers of acinar cell identity (such as amylase) not used to mark acinar cells and metaplastic cells in Figure 2? The authors are currently using different combinations of Cpa1, tdTomato, Krt19 and it is unclear why Cpa1 was chosen over other acinar markers.
4. "7 Krt19+ clusters" are identified in the text referring to Supplemental Figure 3A-C (page 7), but only 6 clusters are shown in that figure.

5. Figure 2J is hard to interpret, given the text that states, "Onecut2 and Foxq1 were expressed in 33.4% and 14.9% of Krt19+ cells, respectively (Figure 2J)."
6. Supplemental Figure 3J should include a p-value.

Reviewer #2 (Remarks to the Author):

In this manuscript, the authors conducted scRNA-seq of pancreatic cancer development from pre-invasive stages to cancer in mouse model, with validation in human PDAC. By dissecting the heterogeneity of metaplastic acinar cells, they discovered two metaplasia-specific transcription factors and proposed that there were different metaplastic cell types in pre-invasive lesions. Among them, metaplastic stomach-specific cell types were detected. Moreover, they characterized the single cell transcriptomes of stromal cells, including immune cells, endothelial cells and fibroblasts, which may support tumor progression.

In summary, this is a descriptive study but the data presented in the manuscript presents a valuable resource. The authors present interesting findings but some of them are short for further experimental validation for backing their claims. Hence, we can only recommend this manuscript for publication in Nature Communications if these shortcomings, listed in detail below, have been addressed.

Major critiques:

1. Cell state classification. Metaplastic cells are heterogeneous as shown by authors. Are they all cells in panIN stages or a mixture of cells in ADM and PanINs stages? PanINs can be classified as low-grade and high-grade. Understanding the different cell subtypes and gene expression pattern between (1) ADM and PanINs; (2) low-grade and high-grade panINs; would provide valuable information to delineate mechanism of tumor development, which need further investigation.
2. Experimental validation and functional exploration. The two metaplasia-specific transcription factors were computationally identified. However, this founding needs experimental validation such as immunostaining in control, metaplastic and tumor tissues. Moreover, some functional exploration of the obtained findings would provide important information.
3. It's nice to investigate the metaplastic stomach-specific cell types and validate these clusters by immunostaining in mouse samples. Immunostaining of stomach-specific cell markers in human samples are also needed to support their findings. In Fig 3F-H, GKN1 positive Pit-like cells located in panIN lesions and the proportions of GKN1-positive cells in ductal or metaplastic structures varied among control, 5 months and 9 months. Nearly most of the cells in ductal structures in 5 months are GKN-1 positive. Is it accord to the proportion of GKN-1 positive cells in computationally analysis? Or just the bias due to the sample preparation and profiling strategy in this study? It's important since their cell trajectory analysis indicates acinar cells could proceed to two different parallel stages: metaplastic stomach-like cells and tumor cells. If most of the metaplastic cells in 5 months are stomach-like cells, it seems that only a small proportion of cells can develop to the tumor cells. While in Fig 4E, the same concern exists that Tgfb-positive cell, represented the senescent state, is also the major cell type in the ductal structures.
4. The authors identify several distinct subpopulations of metaplastic cells. Are different subpopulations mouse-specific? Or they were collected from different mice?
5. With regard to the potential interaction of metaplastic cells with immune populations, this is a somewhat superficial and no-validated hypothesis. Thus, the authors could consider to perform spatial imaging analysis e.g. using specific markers enriched in one metaplastic subtype like pit-like cells and the specific immune cells like neutrophils.
6. With regard to most of the represented IHC figures, statistics are needed.

Minor issues:

1. Labeling order is confusing. It is very difficult to read the figures in order fluently. It is better to adjust figure order for easy following.
2. There is a typo that legend of Fig 6K. It is Agr2, not Arg2.
3. All the selective markers used to define activate fibroblast need to be stated with cited references.

We thank both reviewers for their constructive and helpful comments. We have followed their constructive ideas and useful suggestions to improve the paper. Following the request from reviewer 2 and due to the addition of new panels, figure numbering and legends were updated and reordered.

Reviewer #1 (Remarks to the Author):

Summary

The manuscript entitled ‘Single cell transcriptomes of pancreatic pre-invasive lesions and pancreatic cancer reveal the full heterogeneity of acinar metaplastic cells’ by Schlesinger et al. reveals interesting observations regarding the plasticity of acinar cells during the early stages of Kras-driven transformation in pancreatic cancer. Although a similar analysis has recently been performed by the Wu lab (Peng et al., 2019 Cell Research), the approach taken by the authors of this manuscript led to interesting conclusions that stand distinct. In particular, the previous study focused primarily on building an atlas using human pancreas ductal adenocarcinoma (PDAC) tissue, while the work by Schlesinger and colleagues used tdTomato lineage labeling of acinar cells in a mouse model. Additionally, the authors looked to establish new routes of plasticity within acinar cells, while previous work focused on ductal cell sub-populations. This important detail sheds light on how acinar cells might take different routes to transdifferentiate into ductal-like cells, and therefore represents a significant step forward in our understanding of acinar-to-ductal metaplasia. In summary, the novel approach (fluorescent tagging of acinar cells) is one that can only be performed in mouse models and has gleaned novel insights into cellular plasticity of the pancreas. However, I have the following major and minor concerns:

Major Concerns:

1. *The ‘metaplastic cell’ population, defined by (Cpa1-, tdTomato+, Krt19+) will specifically select for late ADM cells that have completely lost their acinar cell identity (Cpa1-), but what about triple positive (Cpa1+, tdTomato+, Krt19+) cells that would mark acinar cells as they undergo initial ADM. This critical difference might account for why the authors are finding such distinct differences in the cell populations. For a more complete analysis, the authors should examine and discuss earlier ADM cell populations using the above mentioned triple positive cells.*

Response: We greatly appreciate the reviewer’s suggestion, which pointed to an important population of early metaplastic cells which we missed before. Following the review, we have analyzed triple-positive (Cpa1+, tdTomato+, Krt19+) cells and were able to identify two sub-populations of triple positive cells (**new Supplementary Figure 3B**). The first such population was not significantly different from acinar cells, showing no differentially expressed (DE) genes other than *Krt19* (these cells are marked in blue in fig. 3B). However, the second population clustered tightly (**new Supplementary Figure 3C**) and had more than 1000 DE genes compared to acinar cells from non-induced mice (**NEW Supplementary Table 5**). This population also differed from the late metaplastic cells (Cpa1-, tdTomato+, Krt19+) (**NEW Supplementary Table 6**). In the revised paper, we define this new population as early metaplastic. Based on the signatures of acinar, ductal and metaplastic cells, we show that the triple-positive cells have an intermediate transcription profile score, positioned between the metaplastic expression signature and the acinar expression signature (**new Supplementary Figure 3D**).

Interestingly, the early metaplastic cells do not express *Onecut2* and *Foxq1*, which may implicate these two transcription factors as having a role in the transition from an early metaplastic stage to a late metaplastic stage.

We have now added a description of this intermediate population in the results section, as well as the relevant figures and tables.

2. *The authors have defined heterogeneity via scRNA-seq, but the data lack depth. In particular, data discussed (i) on page 6 in regard to Figure 2 and (ii) on page 10 in regard to Figure 5 are lacking. Can the authors directly show any functional importance of the short list of transcription factors (Figure 2) and secreted factors (Figure 5) discussed? Functional data, in vitro and/or in vivo, would strengthen the authors’ conclusions in these key figures. For example, expression of ligand and receptor pairs (Figure*

5) does not necessarily mean that those are the factors truly responsible for immune cell recruitment. Additionally, this data could be strengthened by correlative studies in tissue.

Response: We agree with the reviewer comments and would like to highlight that in addition to defining the heterogeneity using scRNA-seq, we performed extensive validation using immunohistochemistry, and have now added quantification of different metaplastic cell-types in ADM and PanINs based on the immunohistochemistry experiments (**new Figure 4C, Supplementary Figure 7G-I**). These experiments assisted us in exploring the localization of the metaplastic cells and verifying the expression of the relevant protein markers.

We agree with the reviewer's comment that the expression of ligand and matched receptors may not be functional; however, isolating sub-populations of metaplastic cells is technically challenging and these cells are short lived *ex vivo*. We addressed the issues raised in this point in the following ways:

- (i) We performed an analysis to test if cytokines expressed by metaplastic cells affect the potential target cells, based on the receptor expression. We specifically explored the interaction reported in Figure 5, between metaplastic pit-cells that express *Il18* and metaplastic senescent cells and CD4⁺ T cells that express *Il18R1*. We found that Il18 downstream signaling could be detected in metaplastic senescent cells (p-value 0.01) and CD4⁺ T cells (p-value 2.7×10^{-10}) but not in other metaplastic cell-types or CD8⁺ T cells (**new Supplementary Figure 8A-B**) that do not express the receptor. This outcome supports our notion that a downstream program is induced. The new findings support the conclusion that Il18 mediates the induction of TNF, IL1A, Timp1 and other secreted factors by metaplastic senescent cells.
- (ii) We performed correlative studies in tissue, as suggested by the reviewer. These experiments demonstrated co-localization of the metaplastic pit-like cells (in PanINs), neutrophils and T cells. These findings strengthen the possibility that cytokine-receptor interactions presented in figure 5 occur *in-vivo* (**new Supplementary Figure 8C-H**).
- (iii) We explored the potential function of *Onecut2* expression in advanced metaplastic cells. We used CHIP-seq published data of *Onecut2*, obtained from A549 cells¹. We found a significant overlap between *Onecut2* binding sites and genes expressed by metaplastic cells in our study, including enrichment for genes that control Kras downstream signaling (**new Supplementary Figure 3E**). This analysis supports the conclusion that *Onecut2* has a key role in the early events of cellular transformation. In addition, we stained for *Onecut2* in human and mouse tumor samples to verify the expression of this transcription factor at the protein level (**new Supplementary Figure 4K-M**).

Together, the new analyses shed light on the functional importance of the cytokine receptor signaling that we reported and of transcription factors that are uniquely expressed by metaplastic cells.

3. *The discussion is critically lacking in a comparison to demonstrate how this current work is distinct from the work previously published by the Wu lab (Peng et al., 2019 Cell Research), which is relatively easy to do.*

Response: We certainly agree and added a relevant section in the beginning of the discussion.

Minor

Concerns:

1. *More details should be provided on page 6-7 to explain the rationale for narrowing down to the 5 transcription factors discussed in more detail (Figure 2F-H). Are previously known transcription factors also identified?*

Response: Transcription factors that are known to be expressed in metaplastic cells, such as Sox9, are mentioned in the manuscript (Figure 2F). Id1 and Id3 were previously reported as being highly expressed in PDAC samples², a reference which was now added to the revised manuscript. The reason for the narrowing down and focus on these two TFs, is that *Onecut2* and *Foxq1* are uniquely expressed in metaplastic cells (Figure 2I) but *Id1*, *Id3*, and *Runx1* are also expressed in additional cell types based on our data (**NEW Supplementary Figure 3A**).

2. *Supplemental Figure 1L should be expanded to show tdTomato tracking over the different time points of isolation, to potentially show an evolution of the acinar-to-ductal metaplasia event. This might also capture acinar reprogramming events that are occurring.*

Response: We thank the reviewer for raising this point, as the evolution of the acinar-to-ductal metaplasia is one of the central contributions of the paper. We therefore present in Figure 6 a trajectory analysis, in which we explored the gradual changes in tdTomato positive cells over time and found a set of genes that are expressed in early metaplastic cells and can assist to predict if the cell will develop to be a malignant cell or a metaplastic stomach cell. We have added a bar graph showing the changes in the metaplastic cell-type composition over time based on the single-cell data and the counting of metaplastic cell types in ADM and PanINs based on immunohistochemistry staining (**new Figure 4C, Supplementary Figure 7G-I**).

3. *Why are more conventional markers of acinar cell identity (such as amylase) not used to mark acinar cells and metaplastic cells in Figure 2? The authors are currently using different combinations of Cpa1, tdTomato, Krt19 and it is unclear why Cpa1 was chosen over other acinar markers.*

Response: The reduction in the expression of typical acinar enzymes is robust and is not limited to *Cpa1*, as can be seen in Supplementary Table S2 and Supplementary Figure 1 M, N, for *Try4* and for *Amy1*. To show this more clearly, following the reviewer suggestion, we now added a dot plot that includes four enzymes which are highly expressed by acinar cells, *Prss2*, *Amy1*, *Try4* and *Try5* (**new Supplementary Figure 2K**).

4. *“7 Krt19+ clusters” are identified in the text referring to Supplemental Figure 3A-C (page 7), but only 6 clusters are shown in that figure.*

Response: There are seven Krt19+ clusters, H_0, H_2, H_6, H_9, H_11, H_12 and H_13. These clusters are presented in the previous version of Figure 2J. As the reviewer noted, we showed only six clusters in Supplemental Figure 3F. We thank the reviewer for identifying this and have now corrected it and substituted the old Supplemental Figure 3F with a new one, which includes all the seven clusters.

5. *Figure 2J is hard to interpret, given the text that states, “Onecut2 and Foxq1 were expressed in 33.4% and 14.9% of Krt19+ cells, respectively (Figure 2J).”*

Response: Acknowledging that Figure 2J could have been presented in a clearer way, we have prepared a new figure that shows the total percentage of cells that expressed each transcription factor, excluding the sub-division to clusters.

6. *Supplemental Figure 3J should include a p-value.*

Response: The p-value is 0.013 and has now been added to the figure.

Reviewer #2 (Remarks to the Author):

In this manuscript, the authors conducted scRNA-seq of pancreatic cancer development from pre-invasive stages to cancer in mouse model, with validation in human PDAC. By dissecting the heterogeneity of metaplastic acinar cells, they discovered two metaplasia-specific transcription factors

and proposed that there were different metaplastic cell types in pre-invasive lesions. Among them, metaplastic stomach-specific cell types were detected. Moreover, they characterized the single cell transcriptomes of stromal cells, including immune cells, endothelial cells and fibroblasts, which may support tumor progression.

In summary, this is a descriptive study but the data presented in the manuscript presents a valuable resource. The authors present interesting findings but some of them are short for further experimental validation for backing their claims. Hence, we can only recommend this manuscript for publication in *Nature Communications* if these shortcomings, listed in detail below, have been addressed.

Major

critiques:

1. Cell state classification. Metaplastic cells are heterogeneous as shown by authors. Are they all cells in panIN stages or a mixture of cells in ADM and PanINs stages? PanINs can be classified as low-grade and high-grade. Understanding the different cell subtypes and gene expression pattern between (1) ADM and PanINs; (2) low-grade and high-grade panINs; would provide valuable information to delineate mechanism of tumor development, which need further investigation.

Response: We agree with the reviewer that the relationship between the transcription profile, cell identity, and the nature of the lesion is of great importance. The model that we used in our study, expression of *Kras-G12D* in acinar cells, includes a very low frequency of high-grade PanINs³. We have sequenced more than 50,000 single cells but the current setting does not support profiling high-grade PanINs. We also mentioned it in the first section of the results: "It is important to note that nearly all late-stage samples in this model include low-grade PanINs, which are non-invasive."

We could discriminate between ADM and low-grade PanIN lesions and in accordance with the reviewer's main critique, we have added a detailed quantification analysis of metaplastic cell-types and states in ADM and PanINs, based on the immunohistochemistry staining. We counted the number of Tff1 positive cells (pit-like), Mki67 positive cells (metaplastic proliferating), Tgfb1 positive cells (metaplastic senescent), Dclk1 positive cells (tuft-like) and Ins1 positive cells (metaplastic neuroendocrine) in PanINs and ADM structures based on staining with the relevant antibodies. In total we counted more than 42,000 cells from three mice for each antibody (**new Figure 4C, Supplementary Figure 7G-I**). We found that metaplastic tuft-like cells, proliferating cells and senescent cells can be detected in ADM and PanINs and markers of these cell types and states stain both mucin positive and mucin negative cells. Metaplastic pit-like cells however, almost exclusively occupy PanINs and mucin-containing cells.

The metaplastic cell heterogeneity and its profiling is a key finding in the paper. Therefore, the gene expression profile of the lesions (ADM or PanIN) depends on the mixture of metaplastic cell-types in each lesion. The analysis performed in response to the reviewer's comment helped to clarify this issue and better define the composition of metaplastic cell-types in low-grade PanINs and ADM lesions. The relevant text describing the new figures was added to the results and discussion sections of the revised manuscript.

2. Experimental validation and functional exploration. The two metaplasia-specific transcription factors were computationally identified. However, this finding needs experimental validation such as immunostaining in control, metaplastic and tumor tissues. Moreover, some functional exploration of the obtained findings would provide important information.

Response: Following the reviewer comment, we performed immunohistochemistry staining for oncut2 on human and mouse PDAC tumor samples (**new Supplementary Figure 4K-M**). The new staining shows Oncut2 nuclear expression in both human and mouse tumor cells. We have tested several antibodies for FoxQ1 but they did not demonstrate satisfactory performances (positive control of mice stomach tissue sections could not be stained either).

To investigate the function of *Oncut2*, we used published Oncut2 CHIP-seq data from a lung cancer cell line¹. We found a significant overlap of Oncut2 bound promoters with genes that are expressed by metaplastic cells, based on our single-cell data (p-value= 0.00049). The overlapping sets of genes include genes that regulate the RAS/MAPK/ERK pathway. This analysis suggests that Oncut2

functions as a regulator of RAS signaling and cellular transformation. We added the description of the analysis to the revised manuscript (**new Supplementary Figure 3E**).

3. It's nice to investigate the metaplastic stomach-specific cell types and validate these clusters by immunostaining in mouse samples. Immunostaining of stomach-specific cell markers in human samples are also needed to support their findings.

Response: The extent of the similarity between human PDAC development and the mouse cancer model is an important issue. We have extended the analysis of published human PDAC data⁴ to explore if metaplastic cell-types markers that we found in our mice data can be detected in the human PDAC samples. We have found cell clusters that dominantly expressed markers of pit-like cells, chief-like cells, dividing cells and senescent cells, but could not find tuft-like cells (**new Supplementary Figure 7A-B**). These findings are consistent with previous publications that these metaplastic cells were found mainly in early lesions^{5,6}.

In accordance with the reviewer's suggestion, in addition to the new analysis, we stained human PDAC samples that included pre-malignant lesions. We immunostained with anti-Tff1 and anti-Tgfb1 antibodies and could identify positive cells in the tumor tissue and adjacent lesions (**new Supplementary Figure 7C-F**).

Importantly, previous work, which we have cited in our manuscript⁷, reported the expression of stomach markers in bulk RNA-seq of human PanINs and showed staining for Muc6 and Tff1 in human PanINs. Thus, the detailed profiling that we have added, together with previous publications, support the conclusion that our findings are relevant to the development and pathology of human PDAC.

In Fig 3F-H, GKN1 positive Pit-like cells located in panIN lesions and the proportions of GKN1-positive cells in ductal or metaplastic structures varied among control, 5 months and 9 months. Nearly most of the cells in ductal structures in 5 months are GKN-1 positive. Is it accord to the proportion of GKN-1 positive cells in computationally analysis?

Or just the bias due to the sample preparation and profiling strategy in this study?

It's important since their cell trajectory analysis indicates acinar cells could proceed to two different parallel stages: metaplastic stomach-like cells and tumor cells. If most of the metaplastic cells in 5 months are stomach-like cells, it seems that only a small proportion of cells can develop to the tumor cells. While in Fig 4E, the same concern exists that Tgfb-positive cell, represented the senescent state, is also the major cell type in the ductal structures.

Response: We thank the reviewer for raising this important point. The differences in the abundance of Gkn1 positive cells between five months and nine months in Fig 3F-H result from the difference in the lesion types. In this specific figure, in the five-month photo we captured only PanINs, while in the nine month photo we captured a mixture of PanINs and ADM lesions. In both figures, the PanINs are Gkn1 positive and the ductal structures without mucins are Gkn1 negative.

To present an accurate profiling of the mixture of metaplastic cell-types in PanINs and ductal structures, we have now counted more than 42,000 stained cells.

We present three different statistics for each stained protein of metaplastic cell-type or state:

- 1) The number of the positively stained cells in PanINs and ductal structures
- 2) The number of mucin-positive cells and the number of mucin-negative cells
- 3) The number of PanINs that include at least one stained cell of the relevant type and the number of at ductal structures that include at least one stained cell of the relevant type

As detailed above in our response to point 1 of reviewer 2, we found that Tgfb1-positive cells and metaplastic pit-like cells are enriched in PanIN lesions and mucin-positive cells compared to ductal structure and mucin-negative cells. Based on these new data, we could conclude which metaplastic cell types and states occupy more advanced lesions, which shed light on their potential to become malignant. Importantly, although most pit-like cells occupy PanIN lesions, many cells in PanIN lesions (more than 55%), do not express pit cells markers. Thus, we cannot conclude, based on these data alone, what fraction of PanINs cells has the potential to become malignant (**new Figure 4C, Supplementary Figure 7H-I**).

In addition, considering the trajectory analysis, it may still be possible that a small subset of pit-like cells acquire additional mutations and become malignant.

Therefore, at this point it is hard/not possible to determine which metaplastic cell-types develop to malignant cells and the relative amount of cells that have the potential to transform, but our detailed profiling of metaplastic cell-types and states open the way for such discoveries.

Regarding the possible effects of tissue processing, we have quantified the distribution of metaplastic cell-types and states at the different time-points based on the scRNA-seq data (**new Supplementary Figure 7G**), and found that the abundance of the different metaplastic cells types and states do not differ over the late time points (3, 5 and 9 months). Based on the quantitative comparison of the immunostaining and of the single-cell data, tissue processing did not seem to cause significant biases in the distribution of metaplastic cell-types and states.

4. The authors identify several distinct subpopulations of metaplastic cells. Are different subpopulations mouse-specific? Or they were collected from different mice?

Response: The analyses of metaplastic cell types and states are based on three different mice. As detailed above, we now present the full statistical analysis of between-mice variance (**new Supplementary Table 9**).

We have dealt with possible batch effects in the single cell data in the first paragraph of the manuscript. We showed in Supplementary Figure 1J-K that two different mice three months post tamoxifen induction and two different mice five months post tamoxifen induction had a very similar distribution of cell types. In addition, for each cell population, early time points post tamoxifen induction cluster together and late time points post tamoxifen induction cluster together (Figure 1I), showing that batch effects do not affect the profile of cell types, states and differential express genes.

Together, based on the immunohistochemistry and single-cell data from different mice, we have shown that all metaplastic cell-types and states can be detected in each mice in this model, starting three months post tamoxifen induction.

5. With regard to the potential interaction of metaplastic cells with immune populations, this is a somewhat superficial and no-validated hypothesis. Thus, the authors could consider to perform spatial imaging analysis e.g. using specific markers enriched in one metaplastic subtype like pit-like cells and the specific immune cells like neutrophils.

Response: Our data shows that at late time points, simultaneously with PanINs formation, immune cells accumulate in the pancreas.

We have already shown in old Supplementary Figure 5O that T cells localized to PanINs.

In accordance with the reviewer's suggestion we have extended these experiments. We show that T cells and neutrophils can be found in close proximity to metaplastic pit-like cells (**new Supplementary Figure 8C-H**). These experiments strengthen the possibility that the ligand receptors pairs indeed play a role in signaling between different cells in the lesion microenvironment.

Il18 is expressed by metaplastic pit-like cells and the *Il18R1* is expressed by CD4 T cells. We have found that *Il18* downstream signaling is enriched in CD4 T cells but not in CD8 T cells, further strengthening the possibility that the adjacent cells affect each other's function (**new Supplementary Figure 8A**).

6. With regard to most of the represented IHC figures, statistics are needed.

Response: As listed above, we have performed a detailed quantification of each of the stained antibodies sections, based on cell count from three different mice (**new Figure 4C, Supplementary Figure 7H-I, Table 9**). We would like to point to the third point of the responses to this reviewer, where we have described counting more than 42,000 cells and report on whether these cells occupy mucin-positive/negative cells or PanIN/Ductal structures.

Minor

issues:

1. Labeling order is confusing. It is very difficult to read the figures in order fluently. It is better to adjust figure order for easy following.

Response: We have now adjusted the order of the figures to improve the fluency of the reading.

2. There is a typo that legend of Fig 6K. It is Agr2, not Arg2.

Response: We apologize for this typo and have corrected it in the updated version of the manuscript.

3. All the selective markers used to define activate fibroblast need to be stated with cited references.

Response: We added references to the sections describing the endothelial and fibroblast cells.

1. Ma, Q. *et al.* ONECUT2 overexpression promotes RAS-driven lung adenocarcinoma progression. *Sci Rep* **9**, 20021–12 (2019).
2. Maruyama, H. *et al.* Id-1 and Id-2 are overexpressed in pancreatic cancer and in dysplastic lesions in chronic pancreatitis. *The American Journal of Pathology* **155**, 815–822 (1999).
3. Kopp, J. L. *et al.* Identification of Sox9-dependent acinar-to-ductal reprogramming as the principal mechanism for initiation of pancreatic ductal adenocarcinoma. *Cancer Cell* **22**, 737–750 (2012).
4. Peng, J. *et al.* Single-cell RNA-seq highlights intra-tumoral heterogeneity and malignant progression in pancreatic ductal adenocarcinoma. *Cell Res.* **29**, 725–738 (2019).
5. Delgiorno, K. E. *et al.* Identification and manipulation of biliary metaplasia in pancreatic tumors. *Gastroenterology* **146**, 233–44.e5 (2014).
6. Bailey, J. M. *et al.* DCLK1 marks a morphologically distinct subpopulation of cells with stem cell properties in preinvasive pancreatic cancer. *Gastroenterology* **146**, 245–256 (2014).
7. Prasad, N. B. *et al.* Gene expression profiles in pancreatic intraepithelial neoplasia reflect the effects of Hedgehog signaling on pancreatic ductal epithelial cells. *Cancer Res.* **65**, 1619–1626 (2005).

REVIEWERS' COMMENTS:

Reviewer #1 (Remarks to the Author):

The authors have improved their manuscript. While representing a great deal of experiments in the revised manuscript, key functional experiments are lacking to prove causation of the involvement of pathways or transcriptional factors.

Reviewer #2 (Remarks to the Author):

The authors have done a nice work and addressed all the concerns raised by this reviewer. Thanks

Reviewer #1 (Remarks to the Author):

The authors have improved their manuscript. While representing a great deal of experiments in the revised manuscript, key functional experiments are lacking to prove causation of the involvement of pathways or transcriptional factors.

Reviewer #2 (Remarks to the Author):

The authors have done a nice work and addressed all the concerns raised by this reviewer. Thanks

We would like to thank both reviewers for such a quick review, especially in this challenging period. The comments have enabled us to significantly improve the manuscript.

We agree with reviewer #1 that it is very important to understand the function of *Onecut2* and *Foxq1* in greater detail in-vivo and to find the causality. We are therefore making a KO mouse for *Onecut2* and plan to cross it with the mouse model that we used in this study and an additional PDAC mouse model. This work is expected to last for several years and is beyond the scope of this current study.

We have added in the discussion a sentence that is consistent with the reviewer's comment.